# Glacial heterogeneity in Southern Ocean carbon storage abated by fast South Indian deglacial carbon release

Julia Gottschalk [1,2✉], Elisabeth Michel [3], Lena M. Thöle[1,4], Anja S. Studer [5,6], Adam P. Hasenfratz[1,7], Nicole Schmid[1], Martin Butzin [8], Alain Mazaud[3], Alfredo Martínez-García [5], Sönke Szidat[9] & Samuel L. Jaccard [1,10]

Past changes in ocean $^{14}C$ disequilibria have been suggested to reflect the Southern Ocean control on global exogenic carbon cycling. Yet, the volumetric extent of the glacial carbon pool and the deglacial mechanisms contributing to release remineralized carbon, particularly from regions with enhanced mixing today, remain insufficiently constrained. Here, we reconstruct the deglacial ventilation history of the South Indian upwelling hotspot near Kerguelen Island, using high-resolution $^{14}C$-dating of smaller-than-conventional foraminiferal samples and multi-proxy deep-ocean oxygen estimates. We find marked regional differences in Southern Ocean overturning with distinct South Indian fingerprints on (early de-)glacial atmospheric $CO_2$ change. The dissipation of this heterogeneity commenced 14.6 kyr ago, signaling the onset of modern-like, strong South Indian Ocean upwelling, likely promoted by rejuvenated Atlantic overturning. Our findings highlight the South Indian Ocean's capacity to influence atmospheric $CO_2$ levels and amplify the impacts of inter-hemispheric climate variability on global carbon cycling within centuries and millennia.

[1] Institute of Geological Sciences and Oeschger Center for Climate Change Research, University of Bern, Bern, Switzerland. [2] Lamont-Doherty Earth Observatory, Columbia University of the City of New York, Palisades, NY, USA. [3] Laboratoire des Sciences du Climat et de l'Environnement, LSCE/IPSL, CNRS-CEA-UVSQ, Université de Paris-Saclay, Gif-sur-Yvette, France. [4] Department of Earth Sciences, Marine Palynology and Paleoceanography, Utrecht University, Utrecht, Netherlands. [5] Max Planck Institute for Chemistry, Climate Geochemistry Department, Mainz, Germany. [6] Department of Environmental Sciences, University of Basel, Basel, Switzerland. [7] Geological Institute, Department of Earth Sciences, ETH Zurich, Zurich, Switzerland. [8] Alfred-Wegener-Institut Helmholtz-Zentrum für Polar-und Meeresforschung, Bremerhaven, Germany. [9] Department of Chemistry and Biochemistry and Oeschger Centre for Climate Change Research, University of Bern, Bern, Switzerland. [10] Present address: Institute of Earth Sciences, University of Lausanne, Lausanne, Switzerland. ✉email: jgottsch@ldeo.columbia.edu

Past changes in the [14]C inventory of the atmosphere cannot solely be attributed to variations in cosmogenic production (Fig. 1). The fraction of atmospheric [14]C changes unaccounted for by production changes (referred here to as production-corrected $\Delta^{14}C_{atm}$) is believed to reflect large-scale reorganizations of the atmospheric and oceanic respired (i.e. atmosphere-unequilibrated) as well as preformed (i.e. atmosphere-equilibrated) carbon inventories with direct implications for atmospheric $CO_2$ ($CO_{2,atm}$) levels[1,2]. As [14]C is a transient tracer (mean Libby life time $T = 8033$ yr), ocean-versus-atmosphere [14]C disequilibria reflect the rate and efficiency of ocean-atmosphere carbon exchange, and the strength of global-ocean overturning rates[3], and by inference the accumulation of respired carbon in the ocean interior[4]. Ocean [14]C disequilibria are often expressed as ventilation ages that are estimated for instance based on co-existing fossil planktic (surface-dwelling) and benthic (bottom-dwelling) foraminifera, which are believed to faithfully capture the [14]C activity of the water mass they grew in. However, despite a number of [14]C ventilation age reconstructions, important aspects of the past global carbon cycle remain yet unresolved, such as the location and extent of the glacial carbon-rich ([14]C-depleted) reservoir that may explain glacial $CO_{2,atm}$ minima[4,5], the likely carbon transfer pathways during the last deglaciation[5–9], and the rates of oceanic carbon release (uptake) and associated $CO_{2,atm}$ increase (decrease)[4,10].

Intervals of deglacial $CO_{2,atm}$ rise coincide with cold spells in the northern-hemisphere, i.e. Heinrich stadial (HS) 1 and the Younger Dryas (YD), as well as a concomitant rise in Antarctic air temperature (the thermal bipolar seesaw; Fig. 1). Both phases are interrupted by stagnating $CO_{2,atm}$ levels, and warm climate conditions in the North Atlantic, i.e. the Bølling Allerød (BA) interstadial, yet cooling conditions in the Southern Ocean, i.e. the Antarctic Cold Reversal (ACR; Fig. 1). The main control of rising deglacial $CO_{2,atm}$ levels (and concomitant decreasing $^{14}C_{atm}$ levels) on millennial timescales is thought to be the upwelling of deep $CO_2$-rich water masses to the Southern Ocean surface[2,5,7,9]. Southern Ocean upwelling is a highly localized process today that is favored by the interference of the Antarctic Circumpolar Current (ACC) with local bathymetry in regions often referred to as upwelling hotspots[11,12] (Fig. 2). One of these important regions is located in the South Indian Ocean, where the ACC impinges on the Kerguelen Plateau (Fig. 2, Supplementary Fig. 1), causing elevated vertical mixing rates, enhanced cross-frontal exchange and efficient export of subsurface waters to the north[11,12]. Despite the significant leverage of the Kerguelen Island area and similar regions on carbon partitioning between the deep ocean and the atmosphere today (Fig. 2), little is known about their ventilation history and impact on deglacial $CO_{2,atm}$ variations.

Constraints on the past evolution of these upwelling regions can be gained by proxy-based ocean ventilation and oxygenation reconstructions. However, limited carbonate preservation, low abundances of foraminifera and/or uncertainties related to temporal changes in surface-ocean reservoir ages[13] can pose

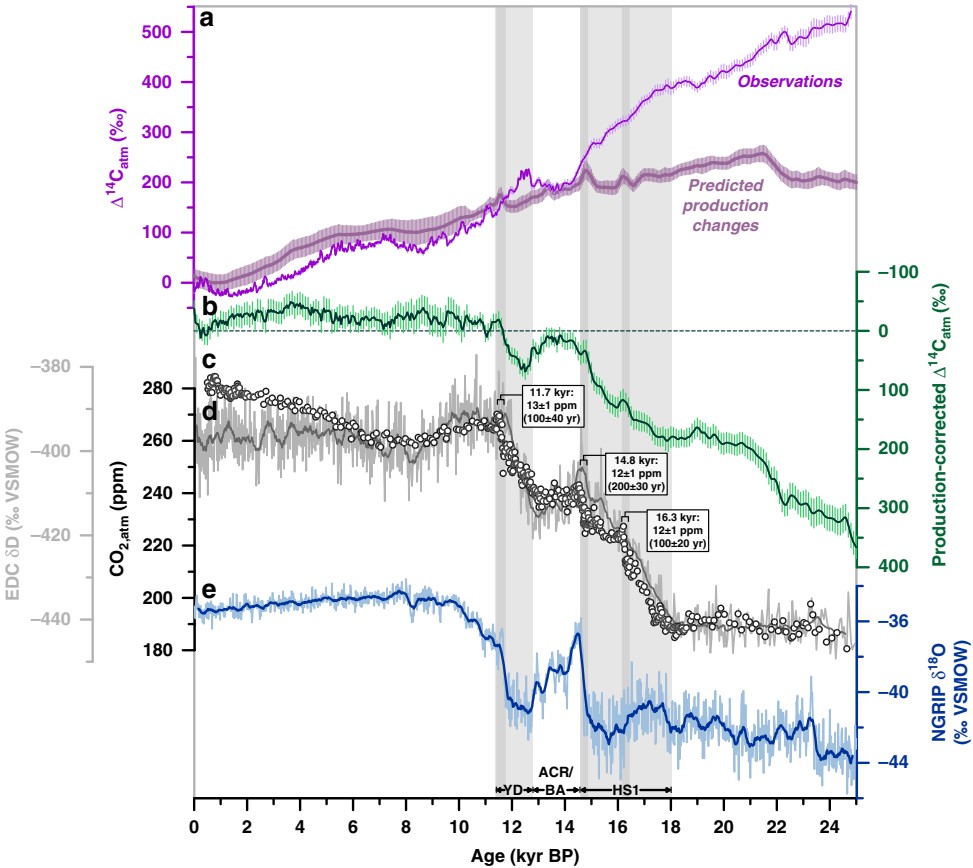

**Fig. 1 Deglacial changes in atmospheric carbon dioxide levels. a** Atmospheric radiocarbon ([14]C) concentrations referenced to modern (i.e. 1950) levels ($\Delta^{14}C_{atm}$, ShCal13, error bars show 1σ-standard deviations (SD))[26] compared to predicted (i.e. modelled) [14]C changes in the atmosphere due to variations in cosmogenic production[3] (with error bars showing 1σ-SD), **b** production-corrected[3] variations in $\Delta^{14}C_{atm}$, with error bars showing 1σ-SD, **c** atmospheric $CO_2$ ($CO_{2,atm}$) variations[32], and **d** Antarctic temperature variations represented by water isotope changes, δD, in the Antarctic EPICA Dome C (EDC) ice core[25], and **e** water isotope changes, δ18O, in Greenland ice core NGRIP[74]. Vertical bars indicate intervals of rising $CO_{2,atm}$ levels. Darker bars highlight intervals of rapidly rising $CO_{2,atm}$ concentrations at ~11.7, ~14.8, and ~16.3 kyr before present (BP)[32]. HS1 Heinrich Stadial 1, ACR Antarctic Cold Reversal, BA Bølling Allerød, YD Younger Dryas.

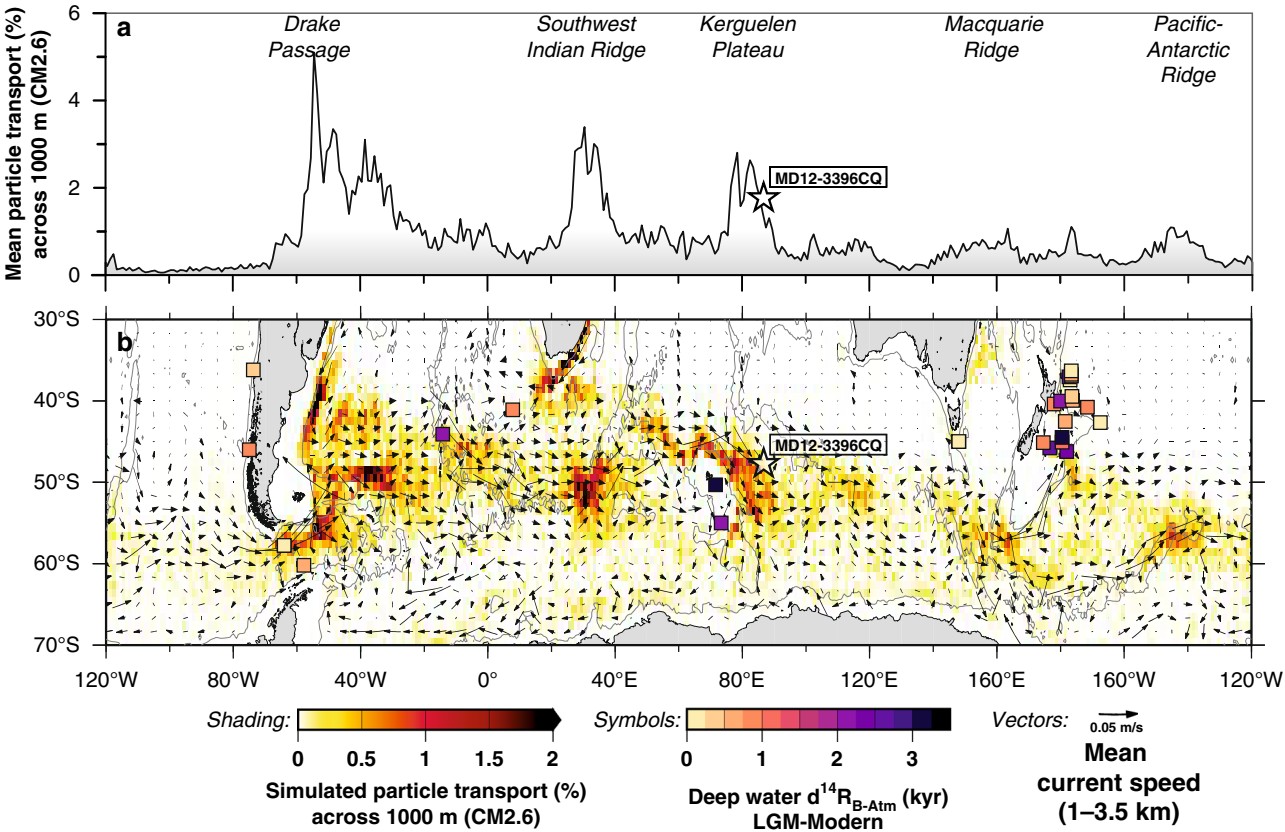

**Fig. 2 Regions of intense interaction of the Antarctic Circumpolar Current with local bathymetry in Southern Ocean upwelling hotspots. a** Zonal variations in the percentage of upwelling particles transport crossing the 1000-m water depth surface (averaged between 30–70°S) as obtained in simulations with the Geophysical Fluid Dynamics Laboratory's Climate Model version 2.6 (CM2.6)[11], where particles were released between 1–3.5 km water depth along 30°S. Increased particle transport in the simulations highlights five major topographic upwelling hotspots in the Southern Ocean[11]. **b** Spatial changes in particle transport in percent across the 1000 m-depth surface, with vectors showing the average speed and direction of ocean currents at mid-depth (1–3.5 km) based on the Global Ocean Data Assimilation System (GODAS) database (https://psl.noaa.gov/data/gridded/data.godas.html) representing the Antarctic Circumpolar Current between 40–60°S. Squares indicate reconstructed deep-water ¹⁴C ages in the Southern Ocean during the last glacial maximum (LGM) referenced to preindustrial[10,16]. Star in both panels shows the location of the study core. Figure modified after ref. [11].

significant challenges to reconstructing deglacial marine carbon cycling in these regions. In addition, sedimentation in these areas can be highly dynamic with common occurrences of high-accumulation drift deposits[14,15], often preventing robust age models to be developed. In particular, stratigraphic alignments of sedimentary iron concentrations to Antarctic ice-core dust as recently employed for the Southwest Indian Ocean[16] may be problematic as the lithogenic fraction may likely not be solely of aeolian origin[17,18]. Many of these limitations were circumvented through paired uranium series- and ¹⁴C-dated corals in the Drake Passage upwelling hotspot region[8] complemented by deep-ocean ¹⁴C ventilation reconstructions from the South Atlantic[5]. The data show a strengthening of upwelling and deep convection in this region from the last ice age until 14.6 kyr before present (BP), which likely contributed to the observed $CO_{2,atm}$ rise during HS1. However, given the challenges and potential shortcomings mentioned above[16], robust estimates of the deglacial evolution of ventilation changes in the deep (South) Indian Ocean are yet limited. While intermediate-ocean ¹⁴C-based ventilation reconstructions offshore the Arabian Peninsula hypothesize two distinct upwelling events in the Indian sector of the Southern Ocean[19], this hypothesis remains untested and ultimately translates into highly uncertain past global carbon budgets associated with the necessity of extrapolations to the deep Indian Ocean[4,10].

Here, we circumvent foraminiferal sample size requirements for most conventional accelerator mass spectrometer (AMS) systems (>1 mg CaCO₃), and reconstruct the deglacial deep-ocean ventilation history of the South Indian Ocean upwelling hotspot east of the Kerguelen Plateau (Fig. 2) via 138 ¹⁴C analyses of small-sized, paired benthic (B) and planktic (P) foraminiferal samples (0.2–1 mg CaCO₃) from sediment core MD12-3396CQ (47°43.88′ S; 86°41.71′ E; 3,615 m water depth; Fig. 2) with the MIni-CArbon-DAting-System (MICADAS) at the University of Bern[20], combined with multi-proxy bottom water oxygen estimates. We show based on multiple lines of evidence that, while the deep South Indian Ocean was a significant (remineralized) carbon sink during the last glacial, marked glacial interbasin differences in carbon storage existed in particular between the Atlantic and Indian sectors of the Southern Ocean, likely due to more weakly ventilated, yet geochemically distinct varieties of Antarctic Bottom Water (AABW). The dissipation of these regional differences was mediated by a reinvigoration of Southern Ocean mixing during the first half of HS1 and enhanced Atlantic overturning at the onset of the BA interstadial, respectively, which we argue promoted a rise in $CO_{2,atm}$ levels. We find that increased Atlantic overturning at the start of the BA period caused a homogenization of the entire South Indian water column, both laterally and vertically, and hence is considered to mark the onset of a modern-like upwelling hotspot near Kerguelen Plateau. Our new findings portray the South Indian Ocean as a more active and distinct player in the global carbon cycle and interhemispheric climate variability during the last deglaciation than previously acknowledged.

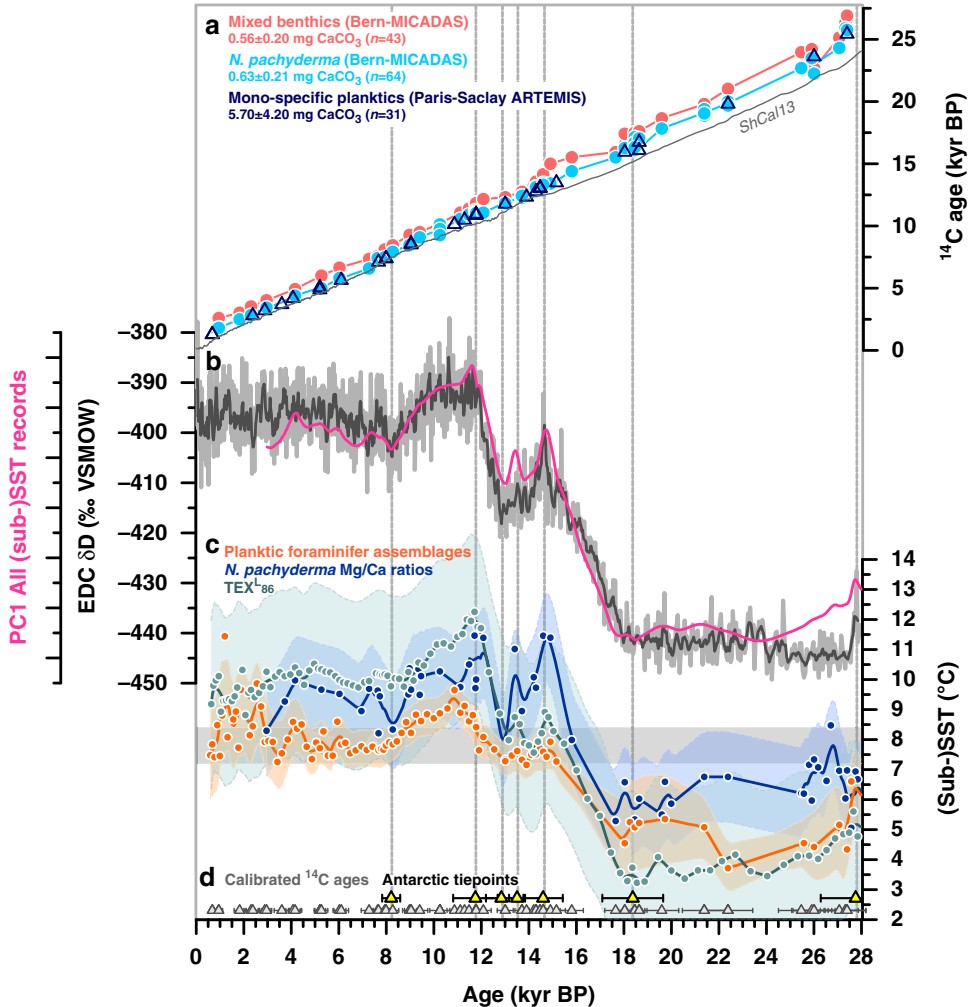

**Fig. 3 Chronostratigraphy and foraminiferal radiocarbon dates in sediment core MD12-3396CQ. a** Benthic foraminiferal (red) and *Neogloboquadrina pachyderma* [14]C dates (light blue) obtained with the Bern-Mini Carbon Dating System (MICADAS, gas and graphite [14]C analyses), as well as planktic foraminiferal [14]C dates obtained with conventional accelerator mass spectrometry (AMS) dating at the ARTEMIS laboratory at the University of Paris-Saclay (open symbols; graphite analyses), grey line shows atmospheric [14]C ages (ShCal13)[26], see Supplementary Fig. 5 for more details, **b** first principal component (PC1) of our three (sub-)sea surface temperature (SST) records (pink) and Antarctica air-temperature variations represented by the EPICA Dome C (EDC) δD record[25] (grey), **c** planktic foraminiferal assemblage-based[21] summer SST changes (orange), TEX[L][86]-based[23] sub-SST estimates (green), and *N. pachyderma* Mg/Ca-based[22] SST variations (blue), envelopes indicate the 1σ standard deviation-uncertainty range (smoothed), and **d** tiepoints between (sub-)SST variations recorded in MD12-3396CQ and δD variations in the EDC ice core[25,69] (yellow and vertical stippled lines, see also Supplementary Figs. 2 and 3), and calibrated planktic foraminiferal [14]C dates (grey). Horizontal bar in **c** indicates the modern SST range at the core site (7.2–8.4 °C, 0–50 m average; World Ocean Atlas 2013)[24].

## Results

**Deglacial variations in upper ocean temperatures.** Our study takes advantage of a comprehensive age model approach based on a stratigraphic alignment of multi-proxy (sub-)sea surface temperature ((sub-)SST) records and Antarctic air temperature as recorded in Antarctic ice cores. We reconstructed (sub-)SST variations at our study site based on three independent foraminiferal and lipid biomarker proxies (i.e. foraminiferal assemblages[21], planktic foraminiferal Mg/Ca ratios[22], and the TEX[L][86]-paleothermometer[23]; Methods). All temperature proxy records closely resemble Antarctic temperature variability (Fig. 3), and reconstructed late Holocene values agree well within uncertainties with the present-day annual SST range at the core site (7.2–8.4 °C, 0–50 m average)[24] (Fig. 3). We assess (sub-)SST variability independently of the uncertainties inherent to each proxy based on the first principal component (PC1) of all three datasets (Fig. 3), which accounts for 77% of the data variance. Under the assumption of thermal equilibrium between sub-Antarctic and

Antarctic temperatures[5], we graphically align PC1 with Antarctic air temperature recorded by δD variations in the Antarctic EPICA Dome C (EDC) ice core[25] (Fig. 3, Supplementary Figs. 2–4). The obtained tiepoints provide estimates of local surface-ocean reservoir age (d[14]R[P-Atm]) variations (Fig. 4, Supplementary Fig. 5). Using these d[14]R[P-Atm] constraints, we correct and calibrate our 95 planktic foraminiferal [14]C dates based on the atmospheric ShCal13 calibration[26], and calculate a sediment deposition (i.e. a chronological) model for our study core (Methods, Supplementary Fig. 6).

**Surface-ocean reservoir age changes.** High-resolution [14]C dates (Methods[20]) obtained from small monospecific planktic foraminiferal samples (*Neogloboquadrina pachyderma*) in gaseous (0.24–1.07 mg CaCO$_3$; $n = 57$) and graphite form (0.63–1.15 mg CaCO$_3$; $n = 7$) show a steady down-core [14]C age increase without age reversals that exceed 2σ-uncertainties (Fig. 3; Supplementary Fig. 5). Gas [14]C analyses of small samples are consistent within

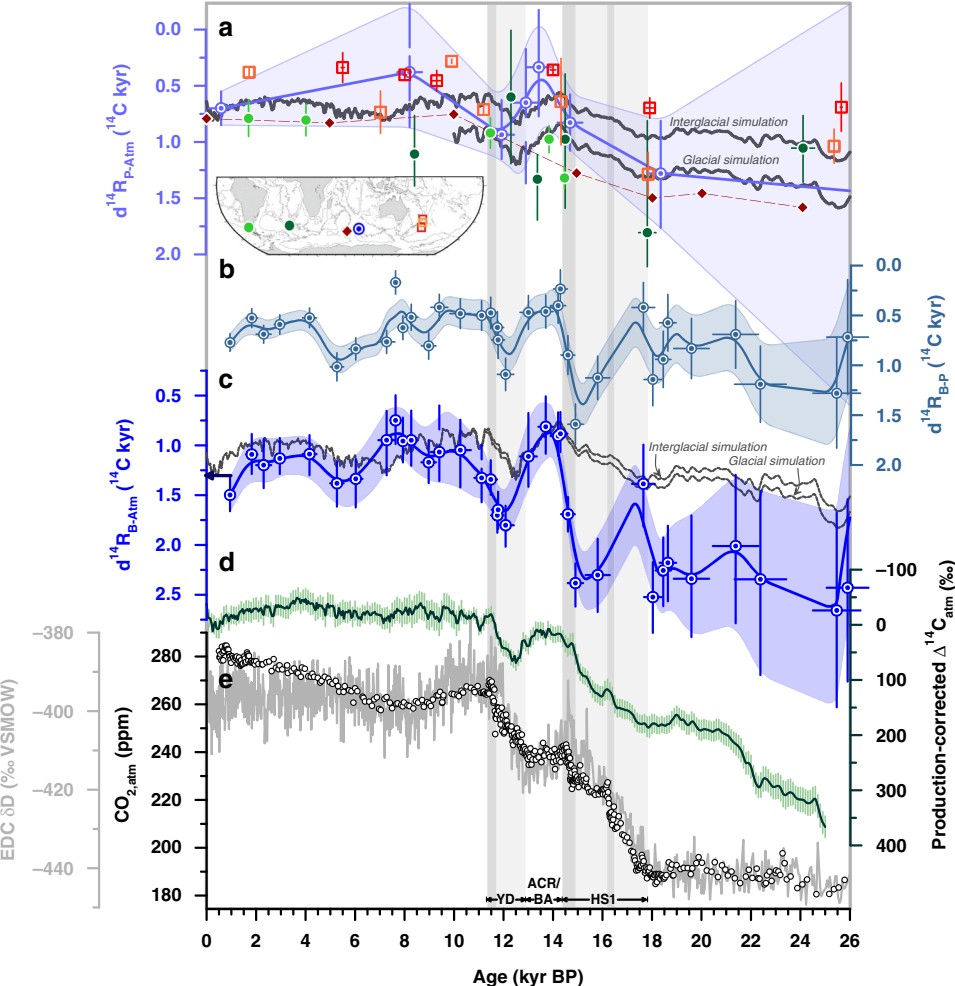

**Fig. 4 Deglacial ocean reservoir age variations reconstructed in South Indian core MD12-3396CQ. a** Surface-ocean reservoir age ($d^{14}R_{P-Atm}$) constraints from the Southern Ocean for the deglacial and glacial periods: Chilean Margin (MD07-3088, light green, tephra-based)[7], in the New Zealand area (orange[30], red[31], tephra-based), in the sub-Antarctic Atlantic (MD07-3076CQ, dark green, stratigraphic alignment between sea surface temperature (SST) and Antarctic temperature)[5], and in the South Indian (MD12-3396CQ, light blue, stratigraphic alignment between (sub-)SST and Antarctic temperature). $d^{14}R_{P-Atm}$ values for the Kerguelen Plateau adopted by ref. [16] are shown in dark red (for locations of cores see inset map), **b** benthic-to-planktic foraminiferal $^{14}C$ age offsets ($d^{14}R_{B-P}$) in MD12-3396CQ, **c** $^{14}C$ age offsets of benthic foraminifera in MD12-3396CQ from the contemporaneous atmosphere, $d^{14}R_{B-Atm}$, **d** $\Delta^{14}C_{atm}$ variations corrected for changes in cosmogenic $^{14}C$ production[3], and **e** atmospheric CO₂ (CO₂,atm) changes (black symbols)[32] and EPICA Dome C (EDC) δD variations (grey)[25,69]. Grey lines in **a** and **c** show simulated $d^{14}R_{P-Atm}$ changes at the study site (80–100°E, 45–50°S; Methods[29]) at 25 m and 3.5 km depth, respectively. Arrow in **c** indicates prebomb deep-ocean reservoir ages (1.3 $^{14}C$ kyr) at our study site (according to the the Global Ocean Data Analysis Project database, version 2)[28]. Lines and envelopes show 1 kyr-running averages and the 1σ-uncertainty/66%-probability range. Vertical bars indicate intervals of rising CO₂,atm levels (darker bands highlight periods with centennial-scale CO₂,atm increases[32]). HS1 Heinrich Stadial 1, ACR Antarctic Cold Reversal, BA Bølling Allerød, YD Younger Dryas.

$100 \pm 70$ $^{14}C$ yr ($n = 7$) with those made on larger graphitized samples (0.59–13.75 mg CaCO₃; $n = 31$), despite a sample mass difference up to a factor of ~8 (Fig. 3; Supplementary Fig. 5)[20].

We estimate $d^{14}R_{P-Atm}$ variations through subtracting atmospheric $^{14}C$ ages, derived from our 10 calendar (cal.) tiepoints, from our interpolated (high-resolution) planktic $^{14}C$ age record (Fig. 4, Supplementary Fig. 5). We find that reconstructed $d^{14}R_{P-Atm}$ ages deviate from preindustrial (i.e. prebomb) surface-ocean reservoir ages of $700 \pm 150$ yr at our study site (Fig. 4, Supplementary Fig. 5)[27,28]. Specifically, $d^{14}R_{P-Atm}$ values during the last glacial and the early/late deglacial warming intervals were elevated by up to 800 and 200 yr, respectively (Fig. 4). In contrast, the Holocene and the brief deglacial period of the ACR show lower $d^{14}R_{P-Atm}$ values by up to 400 yr (Fig. 4). Our proxy data-based constraints on $d^{14}R_{P-Atm}$ are supported by

transient simulations of marine reservoir ages in the study area (80–100°E, 45–50°S; Fig. 4a) that are forced by temporal changes in $\Delta^{14}C_{atm}$ (Methods[29]). Specifically, our glacial reconstructions resemble modelled $d^{14}R_{P-Atm}$ under glacial boundary conditions, while $d^{14}R_{P-Atm}$ values broadly agree with simulations under present-day climate boundary conditions beyond the ACR (Fig. 4a). Although uncertainties in our $d^{14}R_{P-Atm}$ estimates including chronological, analytical and calibration errors (Methods) are nontrivial, changes in surface-ocean reservoir ages at our study site match with proxy-data-based $d^{14}R_{P-Atm}$ reconstructions from the Southwest Pacific[30,31], the Chilean margin[7] and, apart from one mid-glacial datapoint, the South Atlantic[5] (Fig. 4a). In contrast, our reconstructions disagree with the deglacial $d^{14}R_{P-Atm}$ variability recently inferred for the Kerguelen Plateau region[16] on millennial timescales (Fig. 4a).

**Deglacial deep-ocean 14C disequilibria in the South Indian Ocean**. Our study site is bathed by Lower Circumpolar Deep Water (LCDW), underlain by AABW and overlain by Indian Deep Water (IDW, Methods and Supplementary Fig. 1), and is thus ideally located to document the temporal evolution of vertical mixing due to its sensitivity to changes in northern- (emanating from the North Atlantic) and southern-sourced water masses (originating primarily from Adélie Coast and the Ross Sea). This can be achieved by determining past changes in deep-ocean $^{14}$C disequilibria that are reflected in $^{14}$C age offsets between deep water (benthic foraminifera) and the surface ocean (planktic foraminifera) (d$^{14}R_{B-P}$) as well as the contemporaneous atmosphere (d$^{14}R_{B-Atm}$).

We observe slightly higher d$^{14}R_{B-P}$ values at our core site during the last glacial maximum (LGM, i.e. from 23–18 kyr BP) (900 ± 250 yr, $n = 6$) when compared to the Holocene (last 11 kyr BP: 600 ± 200 yr, $n = 14$; Fig. 4b). Rapid decreases in d$^{14}R_{B-P}$ values occur at the end of the early- and late deglacial (Fig. 4b). Consistent with our d$^{14}R_{B-P}$ record, reconstructed d$^{14}R_{B-Atm}$ values are significantly elevated during the LGM (2300 ± 200 yr, $n = 6$) compared to the Holocene (1100 ± 200 yr, $n = 14$) (Fig. 4c). They further show an abrupt d$^{14}R_{B-Atm}$ decrease at the onset of the early deglacial warming and rising CO$_{2,atm}$ levels at ~18.3 kyr BP, but an increase to glacial-like conditions shortly thereafter (Fig. 4c). Although this feature is constrained by only one paired $^{14}$C measurement, it is independently supported by bottom water oxygen reconstructions, as discussed below. The strongest deglacial change in d$^{14}R_{B-Atm}$ occurs at ~14.6 kyr BP, when d$^{14}R_{B-Atm}$ rapidly decreases by 1500 ± 300 $^{14}$C yr within 600 ± 400 cal. yr in parallel with a CO$_{2,atm}$ increase[32] of 12 ± 1 ppm at the onset of the ACR and BA (Figs. 4c, 5). The deep South Indian Ocean remained well-ventilated during the ACR, when d$^{14}R_{B-Atm}$ ages are lower by up to 500 yr than prebomb values (Fig. 4c). Late deglacial (i.e. YD) warming in the southern high latitudes coincides with a rise of d$^{14}R_{B-Atm}$ ages. At ~11.7 kyr BP, however, d$^{14}R_{B-Atm}$ rapidly drops by 450 ± 250 $^{14}$C yr within 850 ± 400 cal. yr, paralleling a marked centennial-scale CO$_{2,atm}$ rise[32] of 13 ± 1 ppm (Fig. 5). Reconstructed mean late Holocene d$^{14}R_{B-Atm}$ values at our core site (1200 ± 200 yr) agree well with prebomb values (Fig. 4c)[28]. Numerical simulations forced by changes in air-sea CO$_2$ exchange through transient changes in Δ$^{14}$C$_{atm}$ and CO$_{2,atm}$ (Methods[29]) also broadly agree with our reconstructed deep-ocean ventilation ages during the late deglaciation and Holocene, while they significantly underestimate deep-ocean reservoir ages during the LGM and early deglaciation (Fig. 4c).

Our deep South Indian ventilation ages closely resemble those found further downstream at mid-depth of the Southwest Pacific[30,33] (1.6–3 km; Fig. 5). In contrast, upstream in the South Atlantic at 3.8 km water depth[5] (a site chosen because of a comparable hydrography and methodological approach used for our study core), glacial d$^{14}R_{B-Atm}$ values are found to be much larger by 300–1500 $^{14}$C yr than at our study site (Fig. 5), which is also reflected in differences in epibenthic δ$^{13}$C and δ$^{18}$O records[34] (Supplementary Fig. 9). However, d$^{14}R_{B-Atm}$ values in both regions converge and decrease simultaneously (within age uncertainties) at ~14.6 kyr BP and with identical magnitude (~1500 $^{14}$C yr) (Fig. 5e, f), and share similar variability thereafter. During the subsequent ACR, reduced ventilation ages in both the deep South Indian and South Atlantic Oceans closely agree with intermediate water estimates from south of Tasmania (1.4–1.9 km)[35] and with the mid-depth Southwest Pacific Ocean (1.6–2.3 km; Fig. 5e, f)[30,33]. A similar convergence can be observed at the end of the late deglacial warming interval, i.e. the end of the YD (Fig. 5c, d).

**Deglacial bottom water oxygenation changes**. The diagenetic precipitation of insoluble (authigenic) U compounds in marine

sediments and in foraminiferal coatings is redox-driven, with oxygen-depleted conditions in pore waters favoring the enrichment of aU (Methods). At our South Indian study site, changes in bulk sedimentary aU levels during the last deglaciation closely parallel reconstructed variations in d$^{14}R_{B-Atm}$ ages (Fig. 6), and are entirely consistent with changes in the enrichment of U compared to Mn in authigenic coatings of foraminifera (Fig. 6).

A more quantitative bottom water [O$_2$] indicator at our study site is provided by the δ$^{13}$C difference between the benthic foraminifera *Globobulimina* sp. and *Cibicides* sp., the Δδ$^{13}$C proxy[36]. This proxy is thought to reflect the oxygen-driven respiration of organic matter in marine subsurface sediments, and is thus a direct measure of bottom water [O$_2$] (Methods[36]). Applying the most recent Δδ$^{13}$C-[O$_2$] calibration[36], we find that bottom water [O$_2$] during the LGM was lowered by 100 ± 40 μmol kg$^{-1}$ from present-day concentrations (~220 μmol kg$^{-1}$; Fig. 6)[37], which is consistent with higher d$^{14}R_{B-Atm}$ values, increased sedimentary aU enrichments and higher foraminiferal U/Mn ratios (Fig. 6). In contrast to our d$^{14}R_{B-Atm}$- and bulk sedimentary aU records, we do not observe a marked Δδ$^{13}$C-based bottom water [O$_2$] change during the early deglaciation, suggesting that any bottom water [O$_2$] change during this interval must have been confined to within the proxy uncertainty, i.e. 80 μmol kg$^{-1}$, if it existed at all. Our foraminiferal Δδ$^{13}$C data additionally highlight a rapid bottom water [O$_2$] increase of 50 ± 40 μmol kg$^{-1}$ at the end of the early deglacial warming at 14.6 kyr BP in parallel with the rapid reductions in d$^{14}R_{B-Atm}$, aU levels and foraminiferal U/Mn ratios (Fig. 6).

## Discussion

**Enhanced deep South Indian Ocean carbon storage during the LGM**. An isolated glacial deep-ocean $^{14}$C-depleted carbon reservoir that accommodated more respired carbon during the LGM has been proposed to explain the last glacial CO$_{2,atm}$ minimum[4]. Marine $^{14}$C proxy evidence supports the existence of such a reservoir in the Pacific Ocean[30,38], in the deep South Atlantic[5] and in the deep North Atlantic[39]. Larger-than-Holocene d$^{14}R_{B-Atm}$ (and d$^{14}R_{P-Atm}$) values (Fig. 4), lower Δδ$^{13}$C-derived glacial bottom water [O$_2$] and enhanced glacial aU accumulation at our study site (Fig. 6) extend those observations to the Indian sector of the Southern Ocean, which is consistent with recent findings from the Crozet and Kerguelen Plateau regions[16]. Assuming negligible saturation- and/or disequilibrium [O$_2$] changes, our multi-proxy reconstruction highlights a substantially higher accumulation of respired carbon throughout the deep Indian Ocean basin during the LGM than during the Holocene. The last glacial ventilation age increase at our South Indian core site is about twice that of the global LGM ocean mean[4,10] (Fig. 4), similar to observed trends in the Atlantic[5] and Pacific[33] sectors of the Southern Ocean (Fig. 5). We thus argue that large swaths of the Southern Ocean accommodated an above-average share of the global-ocean respired carbon at the LGM, largely contributing to the observed glacial reduction in CO$_{2,atm}$ through decreased upwelling and ocean-atmosphere CO$_2$ exchange.

The observed ventilation age and Δδ$^{13}$C-derived oxygen changes of glacial South Indian deep waters at our study site are unlike any preindustrial water mass of the deep Atlantic and Indian Oceans, but instead resemble the oldest prebomb deep-water masses of the Pacific Ocean (Fig. 7). As North Atlantic Deep Water (NADW) flow to the glacial Indian Ocean was diminished[40], the glacial (South) Indian increase in respired carbon storage may be related to reduced formation and/or overturning rate of southern-sourced water masses (i.e. AABW) and/or reduced air-sea CO$_2$ equilibration in the Southern Ocean during the LGM. Adjustments of AABW were likely closely

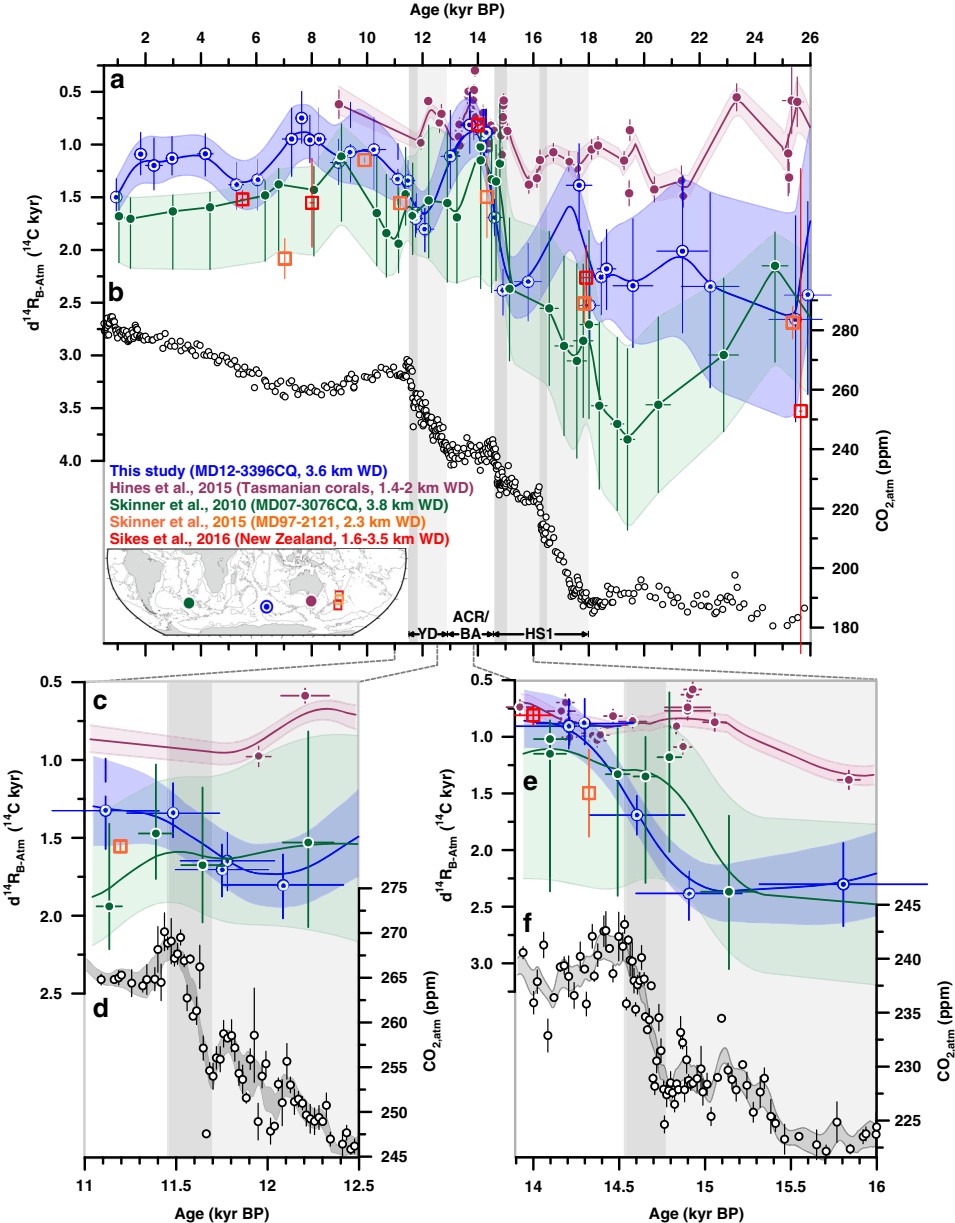

**Fig. 5 Deglacial deep-ocean reservoir age variations in the Southern Ocean. a**, **c**, **e** Benthic foraminiferal $^{14}$C age offsets in MD12-3396CQ (3.6 km water depth, WD) from the contemporaneous atmosphere, $d^{14}R_{B-Atm}$ (blue), compared to deep-ocean ventilation ages in the deep sub-Antarctic Atlantic (MD07-3076CQ, 3.8 km WD; upstream, green[5]), on the Chatham Rise (MD97-2121, 2.3 km WD; downstream, orange[30]), in the New Zealand area (1.6-3.5 km WD; downstream, red[33]), and south of Tasmania (corals, 1.4-1.9 km WD; downstream, purple[35]), and **b**, **d**, **f** atmospheric $CO_2$ ($CO_{2,atm}$) changes. Lower panels zoom in on $d^{14}R_{B-Atm}$ variations during specific intervals of rapid centennial $CO_{2,atm}$ increase[32], i.e. the ~11.7 (**c**, **d**) and ~14.8 kyr events (**e**, **f**). Lines and envelopes show 1 kyr- (**a**) and 0.5 kyr- (**c-f**) running averages and 1σ-uncertainty-/66%-probability ranges. Vertical bars indicate intervals of rising $CO_{2,atm}$ levels (dark bands highlight periods with centennial-scale $CO_{2,atm}$ increases)[32]. HS1 Heinrich Stadial 1, ACR Antarctic Cold Reversal, BA Bølling Allerød, YD Younger Dryas.

linked to a reduction in shelf space around Antarctica owing to advancements of Antarctic ice sheet grounding lines[41], an expansion of Antarctic sea ice cover[42], and an associated shift of the mode and locus of AABW formation, from super-cooling underneath shelf ice and brine rejection in polynyas[43] to open-ocean convection off the shelf break during the last glacial[44,45]. However, open-ocean convection during the LGM may have been localized and seasonal in nature (possibly facilitated by open-ocean polynyas) in all sectors of the Southern Ocean, and therefore less efficient at ventilating the ocean interior. Physical and biological changes in the South Indian Ocean are crucial in explaining the last glacial increase in respired carbon content, because simulated adjustments in air-sea gas exchange alone cannot explain our proxy data (Fig. 4). Therefore, a possible northward shift of the southern-hemisphere westerly wind belt[46] and northward expansion of Antarctic sea ice may have changed the geometry of Southern Ocean density surfaces[47], in particular relative to the location of the Kerguelen Plateau, which combined with more poorly ventilated AABW may have curtailed the capacity of the Kerguelen Plateau region to act as a hotspot for the upwelling of $CO_2$-rich water masses to the surface and subsequent air-sea equilibration.

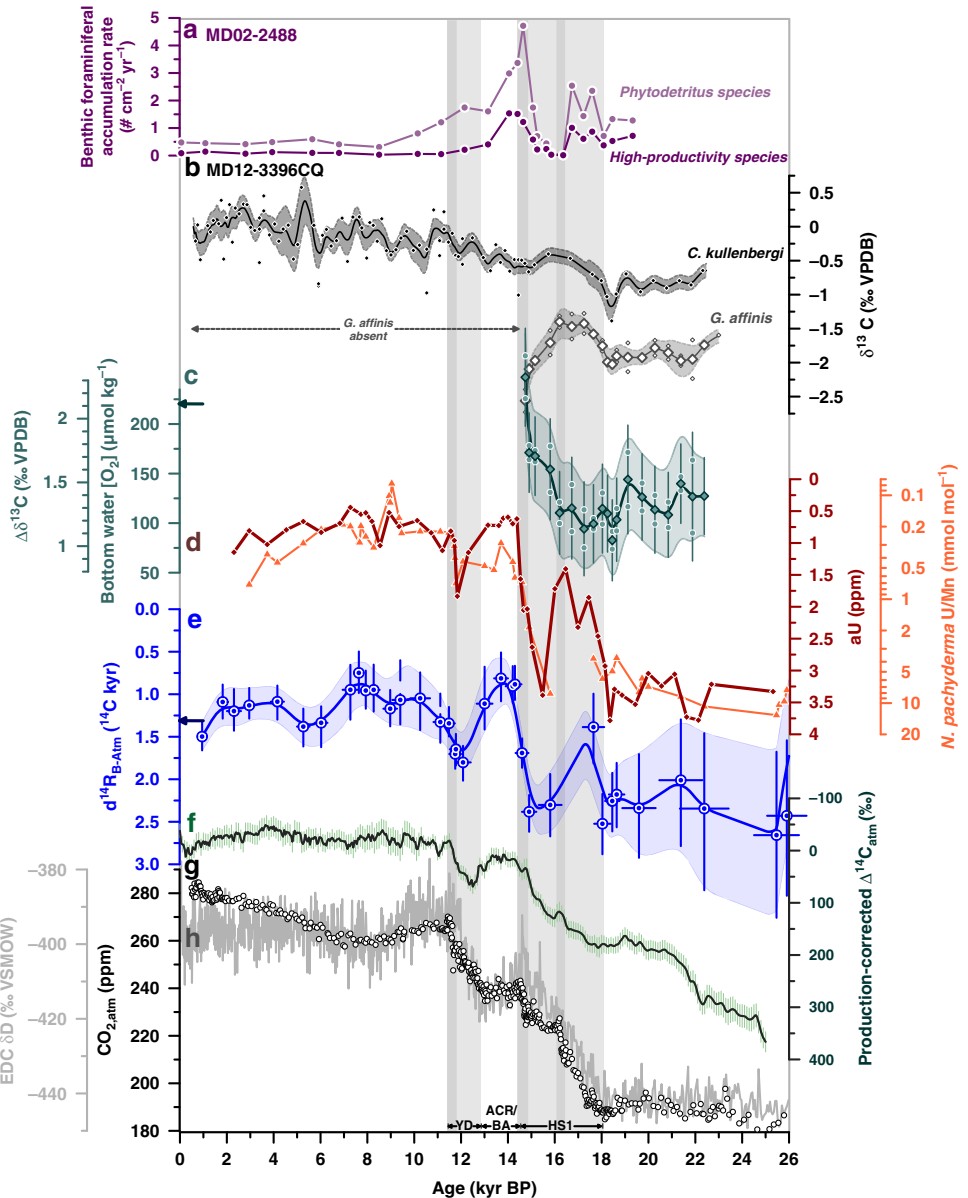

**Fig. 6 Deglacial oxygenation and deep-ocean reservoir age variations in the South Indian Ocean. a** Accumulation rates of benthic foraminifera indicative of phytodetrital input (light purple) and high annual productivity (dark purple) in core[55] MD02-2488, **b** benthic foraminiferal δ[13]C records from MD12-3396CQ (black (epibenthic/shallow infaunal species): *Cibicides kullenbergi*, grey (deep infaunal species): *Globobulimina affinis*; small symbols: replicate analyses, large symbols: mean values), **c** δ[13]C gradient between *G. affinis* and *C. kullenbergi* (Δδ[13]C), and corresponding[36] bottom water [O₂] levels at our study site (arrow indicates present-day bottom water [O₂][37]; small circles show the Δδ[13]C range based on non-averaged *G. affinis* δ[13]C values), diamonds show average values, **d** authigenic U (aU) levels (brown) and U/Mn ratios in authigenic coatings of *N. pachyderma* (orange) in MD12-3396CQ, **e** d[14]R$_{B-Atm}$ variations (arrow shows prebomb values, following the Global Ocean Data Analysis Project database, version 2)[28] measured in core MD12-3396CQ, **f** production-corrected[3] variations in Δ[14]C$_{atm}$, **g** atmospheric CO₂ (CO$_{2,atm}$) variations (circles)[32], and **h** EPICA Dome C (EDC) δD changes (grey line)[25,69]. Vertical bars indicate intervals of rising CO$_{2,atm}$ levels. Darker bands highlight periods with centennial-scale CO$_{2,atm}$ increases[32]. Lines and envelopes in **b**, **c** and **e** show 0.5 kyr-running averages and the 1σ-uncertainty/66%-probability range, respectively. HS1 Heinrich Stadial 1, ACR Antarctic Cold Reversal, BA Bølling Allerød, YD Younger Dryas.

**Glacial-ocean heterogeneities in Southern Ocean carbon storage**. Comparison of our new data with existing paleoceanographic reconstructions suggests common water mass characteristics in the Indian and Pacific sectors of the Southern Ocean during the LGM, yet significant offsets exist with the deep central South Atlantic (Fig. 5). We can rule out that this observed LGM offset is a methodological artifact. First, a consideration of foraminiferal blanks in the reconstruction of d[14]R$_{B-Atm}$ at our study site, although critically discussed[20], can only explain a small fraction of the observed glacial difference (Supplementary Fig. 7).

Second, glacial d[14]R$_{P-Atm}$ estimates in South Atlantic core MD07-3076CQ reach values larger than 2000 years during the LGM[5], which may be unrealistic according to a new compilation[48]. Disregarding these extreme d[14]R$_{P-Atm}$ values (following similar sensitivity tests made by ref. [8]) reduces but does not eradicate the observed LGM d[14]R$_{B-Atm}$ mismatch with our South Indian study site (Supplementary Fig. 8). Further supported by glacial offsets in benthic foraminiferal stable isotopes[34] (Supplementary Fig. 9), we consider the observed interbasin heterogeneity in both d[14]R$_{B-Atm}$ and Δδ[13]C-derived [O₂] values (Fig. 7) to be realistic, and argue

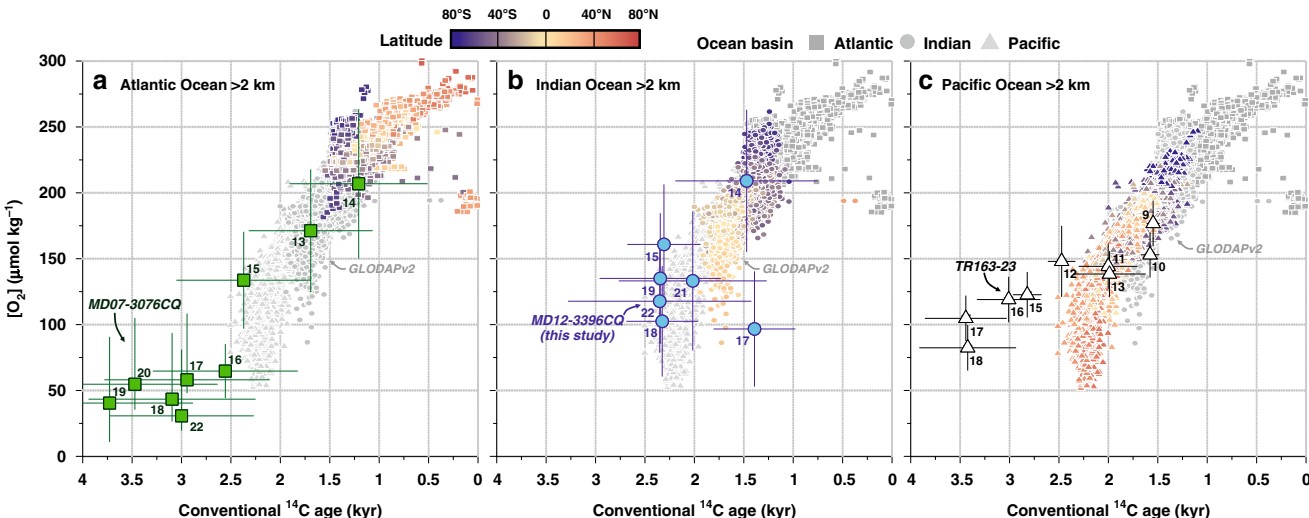

**Fig. 7 Relationship between seawater oxygen concentrations and conventional radiocarbon ages at present-day and in the past.** Modern seawater [O$_2$] levels versus conventional $^{14}$C age in **a** the Atlantic Ocean (squares), **b** Indian Ocean (circles), and **c** Pacific Ocean (triangles) below 2 km water depth[28]; modified after ref. [4]. Symbol color represents the latitude of the seawater sample. Large symbols show reconstructed bottom water [O$_2$] (via the $\Delta\delta^{13}$C proxy) and ventilation ages (i.e. d$^{14}$R$_{B-Atm}$, representing paleo-conventional $^{14}$C ages) from the deep South Atlantic (green: MD07-3076CQ, 3.8-km water depth)[5,6], the deep South Indian (blue, this study: MD12-3396CQ, 3.6-km water depth) and the deep Eastern Equatorial Pacific Ocean (black: sediment core TR163-23, 2.7 km water depth[72,73]; please note that Holocene $\Delta\delta^{13}$C proxy data in this core overestimate present-day bottom water [O$_2$] in the study region by ~80 µmol kg$^{-1}$). Symbol labels indicate temporal bins over which the paleo-$^{14}$C-[O$_2$] data were averaged (in kyr before present (BP), e.g. for 15 kyr BP: 15.99–15 kyr BP). The principal trend of increasing ventilation ages with decreasing seawater oxygen content can be ascribed to the accumulation of respired carbon, while deviations from this trend can be driven by the advection of well-ventilated water masses, e.g. from the Weddell Sea ([O$_2$] increase without $^{14}$C change), or through organic carbon respiration in upwelling regions ([O$_2$] decrease without $^{14}$C change)[4]. On multi-millennial timescales, the respiration rate may change (causing the $^{14}$C-[O$_2$] slope to steepen or flatten), and the ocean-atmosphere $^{14}$C and O$_2$ equilibration timescales change with varying atmospheric CO$_2$ levels (i.e. mean reservoir ages increase in a glacial 190 ppm-CO$_2$ atmosphere without [O$_2$] change)[50] and ocean temperature/salinity (i.e. [O$_2$] saturation increases without $^{14}$C change during glacials)[51].

that they may be explained by distinct sources, modes of formation and/or end-member characteristics of southern-sourced water masses in the glacial Southern Ocean. Present-day differences in the location of AABW formation across the three sectors of the Southern Ocean[43,49] might have been amplified during the LGM owing to reduced NADW inflow[40] and concomitant expansion of AABW[5]. These deep-ocean processes are likely insufficiently captured in the simulations of glacial deep-ocean reservoir ages of ref. [29], in particular because of their dependence on subgrid processes (Fig. 4). As strong regional differences in the surface $^{14}$C equilibration timescale[50], the surface-ocean O$_2$ solubility[51], and respiration rates in the glacial Southern Ocean are unlikely, we conclude that the observed South Atlantic–South Indian offsets in LGM bottom water characteristics may instead be explained by differences in AABW formation and in the degree of ocean-atmosphere CO$_2$ equilibration[44].

The glacial ocean stored an estimated surplus of ~700–1000 GtC, which may explain ~70–100% of the observed glacial-to-interglacial CO$_{2,atm}$ change[4,10,52], although there remains substantial uncertainty[53] regarding the magnitude of terrestrial biosphere loss during the last ice age. However, these estimates required extrapolations to the Indian Ocean basin, as direct observations were lacking. Converting the observed $\Delta\delta^{13}$C-derived LGM-Holocene [O$_2$] offset of 100 ± 40 µmol kg$^{-1}$ at our South Indian study site into respired carbon change following existing protocols[6,52], we find an increase in remineralized carbon level of 55–115 µmol kg$^{-1}$ (Fig. 6, Methods). This is lower or at the lower end of estimates from the South Atlantic (116–144 µmol kg$^{-1}$)[6] and the equatorial Pacific (98–119 µmol kg$^{-1}$)[52], which corroborates our earlier assertion of spatial nuances in the glacial-ocean respired carbon increase across ocean basins. These nuances need to be considered in order to robustly quantify the ocean contribution to glacial CO$_{2,atm}$ minima.

**Transient early deglacial ventilation increase in the South Indian Ocean.** At the onset of HS1, deep-ocean $^{14}$C ventilation (Fig. 5) and oxygenation in the South Indian (as indicated by the aU record in MD12-3396CQ, Fig. 6) and in the South Atlantic[5] increased until 16.3 kyr BP, signaling a net loss of (remineralized) carbon from the deep ocean during that time interval. A comparison to simulated deep-ocean reservoir changes in our study region implies that atmospheric $^{14}$C variability transmitted into the deep ocean cannot explain the observations (Fig. 4). Instead, this early deglacial ocean-carbon release was likely driven through an early deglacial reduction in Antarctic sea ice cover (promoting air-sea gas exchange)[5], increased vertical mixing through AABW reinvigoration[54] and -formation below ice shelves and within coastal/shelf polynyas (as shelf space becomes available), and/or a poleward shift/reinvigoration of the southern-hemisphere westerlies and associated Ekman pumping of subsurface waters[9]. These combined or in isolation might have altered the water column density structure near Kerguelen Plateau in such way that isopycnals increasingly interfered with the local bathymetry, leading to reinvigorated vertical mixing[11,12,47] during the first half of HS1.

We hypothesize that increased South Indian vertical mixing during early HS1 would have fueled biological productivity through the supply of nutrient-rich water masses to the euphotic zone of our study region[17]. This is documented in nearby core MD02-2488 (46°28.8′S, 88°01.3′E; 3420 m water depth) by enhanced accumulation of benthic foraminiferal species indicative of highly seasonal and high surface-ocean productivity as these benthics feed on labile organic matter raining out of the euphotic zone (Fig. 6)[55]. A transient supply of labile phytodetrital organic carbon to the sediment may also help explain the insensitivity of the $\Delta\delta^{13}$C-[O$_2$] proxy as a result of ecological biases[56] of *C. kullenbergi* and/or *G. affinis* during that time

(Fig. 6). Nonetheless, the observed increase in ventilation (via biology-independent paleo-indicators), marine productivity (via benthic foraminiferal response) and in bottom water oxygenation (via aU) during early HS1 provide strong evidence for enhanced upwelling of $CO_2$-rich water masses and/or strengthened upper ocean-atmosphere $CO_2$ equilibration in the South Indian Ocean, which contributed to the early deglacial rise in $CO_{2,atm}$ levels ~18.3 to ~16.3 kyr ago.

A unique feature of the deep South Indian Ocean amongst other existing ($^{14}$C and $O_2$) ventilation age records from south of 40°S relates to a ventilation decrease, and thus a return to glacial-like conditions during late HS1, starting at ~16.3 kyr BP (Fig. 6). Our data signal a restratification of the South Indian water column during late HS1, which may be controlled by changes in the ventilation and formation of South Indian deep waters, for instance by increases in AABW salinity, along with a shift of the southern-hemisphere westerlies to south of the Kerguelen Plateau[57]. A reduction in carbon release from the South Indian to the atmosphere during late HS1, likely due to an unfavorable superposition of the South Indian water column density structure with bathymetry around Kerguelen Island, may have halted the early deglacial $CO_{2,atm}$ rise and promoted the plateauing of $CO_{2,atm}$ between ~16.3 and ~14.8 kyr BP (Fig. 4)[32]. The fact that this late HS1 return to glacial-like conditions in $^{14}$C and $O_2$ ventilation is not observed in the South Atlantic[5,6] or elsewhere in the Southern Ocean indicates that interbasin differences persisted throughout HS1, attributing the South Indian Ocean a unique role in modulating deglacial $CO_{2,atm}$ variability.

**Flushing of the Southern Ocean carbon pool through AMOC reinvigoration.** The marked glacial and early deglacial geochemical interbasin heterogeneity abruptly dissipated at the end of HS1, when the ventilation age and oxygen characteristics of deep waters in the different ocean basins rapidly approached prebomb values for the first time throughout the deglaciation (Fig. 7). We argue that the fast increase in $^{14}$C and $O_2$ ventilation in the South Indian Ocean at the end of HS1, at ~14.6 kyr BP, is linked to a rapid resumption of Atlantic overturning at the onset of the BA warm period[58], causing a rapid (decadal to centennial-scale) southward expansion[59] and eastward deflection of NADW towards the Indian Ocean via the ACC. This may have caused a flushing of the deep-ocean carbon pool that is not only limited to the equatorial Atlantic[60] but expanded into the South Atlantic[5,8] and South Indian Ocean (this study), with remarkable, near-identical ventilation changes in the latter two regions and a much wider spatial impact on the global ocean than previously recognized (Fig. 5a). These flushing events were possibly amplified by a lagged response of sea ice to a rapid shift of the southern-hemisphere westerlies northward[57,61], allowing an unabated, transient evasion of carbon from the Southern Ocean to the atmosphere and likely more efficient Ekman pumping around Kerguelen Plateau. These changes might have also impacted the geometry of Southern Ocean isopycnals relative to the location of the Kerguelen Plateau or underlying bathymetry, causing elevated mesoscale eddy activity, diapycnal exchange and/or upwelling along isopycnal surfaces in our study area[11]. An associated transient upwelling event of $CO_2$ and nutrient-rich water masses to the surface ocean at the BA onset is supported by a second abrupt local abundance peak of benthic foraminiferal species that reflects upwelling-driven phytoplankton blooms (Fig. 6). Our data hence suggest that a large fraction of the concomitant 12 ± 1 ppm $CO_{2,atm}$ increase was driven by a rapid loss of carbon from the South Indian Ocean associated with the reinvigoration of Atlantic overturning and wind-driven Ekman pumping. This transient upwelling event also heralded the establishment of the

South Indian upwelling hotspot at ~14.6 kyr BP that was akin, if not stronger than its present-day counterpart.

A new equilibrium in air-sea gas exchange, however, was reached during the subsequent ACR period, when vertical mixing in the southern, high latitudes was seemingly stronger than at present-day as shown by lower-than-prebomb ventilation ages in the deep South Atlantic[5], in the South Indian (this study) and in the Southwest Pacific[33] (Fig. 5a). Because our South Indian ventilation age reconstructions closely match similar data from the deep South Atlantic[5], the Southwest Pacific[30,33] and intermediate water depths south of Tasmania[35] during the ACR (Fig. 5, Supplementary Fig. 9), we argue that reinvigorated deep-ocean ventilation led to a remarkably homogeneous water column in large parts of the Southern Ocean both laterally and vertically during that time. Given an expansion of Antarctic sea ice cover[61] and northern-hemisphere warming reducing the ocean $CO_2$ solubility[62] during the ACR, the capacity of the South Indian Ocean to impact $CO_{2,atm}$ levels during that time was likely limited despite high mixing rates, which is consistent with the observed $CO_{2,atm}$ plateau during the ACR (Fig. 5a, b).

**Late deglacial South Indian ventilation decrease and early Holocene convergence.** During the YD, we observe decreased $^{14}$C and $O_2$ ventilation at our deep South Indian study site, suggesting that the South Indian remained a moderate source of carbon to the atmosphere, and thus did not significantly contribute to the late deglacial $CO_{2,atm}$ increase. This agrees with inferences made in other Southern Ocean regions[5,7,8], but disagrees with recent findings from the South Indian Ocean[16]. We show that the disagreement results from insufficiently accounted surface-ocean reservoir age variability that can be improved by applying consistent surface-ocean reservoir ages for all sites (Supplementary Fig. 10). At the end of the YD, we observe a convergence of ventilation ages of different parts of the Southern Ocean, reminiscent of the 14.6 kyr BP-event (Fig. 5, Supplementary Fig. 9). We argue that this convergence may have caused a large fraction of the 13 ± 1 ppm-$CO_{2,atm}$ increase at that time through a rapid loss of carbon from the South Indian Ocean mediated by stronger Atlantic overturning and wind-driven Ekman pumping. This reinforces the role of the South Indian Ocean in centennial-scale $CO_{2,atm}$ variability during the last deglaciation.

Based on our high-resolution multi-proxy analyses, we identify marked impacts of marine carbon cycling in the South Indian Ocean on glacial and deglacial $CO_{2,atm}$ variations. While our new high-resolution data support previous contentions of reduced vertical mixing and restricted air-sea gas exchange in all sectors of the Southern Ocean during the last ice age, they point to marked spatial differences, possibly due to lateral variations in the mode and rate of AABW formation and/or the efficiency of air-sea gas equilibration. We argue that major increases in South Indian convection during the early HS1 as well as the ends of the YD- and HS1 stadials as shown by both $^{14}$C (d$^{14}R_{B-Atm}$) and $O_2$ ventilation proxies (aU, $\Delta\delta^{13}$C and foraminiferal U/Mn) affected deglacial $CO_{2,atm}$ variability significantly and rapidly. However, these changes alone cannot explain reconstructed $^{14}$C ventilation ages in the mid-depth North Indian Ocean off the Arabian peninsula during deglacial northern-hemisphere stadials[19].

Glacial and early deglacial spatial differences in deep-ocean ventilation and oxygenation were entirely eroded along with the southward expansion of northern-sourced water masses at the onset of the ACR, with potentially strong effects on $CO_{2,atm}$ levels owing to the flushing of old carbon from the deep ocean including the Indian Ocean. We interpret this to mark the onset of the upwelling dynamics akin to the present-day hotspot region

in our study area. The Indian sector of the Southern Ocean, in particular the upwelling hotspot around Kerguelen Plateau, is thus highly sensitive to northern- and southern-hemisphere climate variability and effective in contributing to deglacial $CO_{2,atm}$ change, which needs to be accounted for in global carbon cycle budgets over centennial, millennial and glacial-interglacial timescales.

## Methods

**Study area.** Sediment core MD12-3396CQ was retrieved from the deep Australian-Antarctic Basin (AAB), south of the Southeast Indian Ridge and east of Kerguelen Plateau (Fig. 2), and is located slightly to the north of the Sub-Antarctic Front (47°43.88′ S, 86°41.71′ E; 3615 m water depth)[63]. The core site is currently bathed in LCDW, which represents the ACC layer influenced by NADW entering the Indian Ocean from the west[64]. Along with underlying AABW, LCDW forms IDW and Upper Circumpolar Deep Water (UCDW) through diapycnal diffusion in the North Indian Ocean[64], which subsequently flows south- and upward to shallower water levels to outcrop south of the Polar Front. In the deep Indian Ocean, AABW enters the basin in the west (originating from the Weddell Sea) filling the deep basins west of the Kerguelen Plateau, and from the east (originating from the Ross Sea and Adélie Coast) filling the deep Australian-Antarctic Basin to the southeast of the study site (Supplementary Fig. 1).

The core is characterized by high sedimentation rates (mostly >10 cm kyr⁻¹), and good preservation of foraminifera. It was recovered from a geographically stable drift deposit south of the Southeast Indian Ridge[14] (Supplementary Fig. 1). This contourite deposit receives detrital material from two sources, in addition to an influence from biogenic fluxes from above. First, the alleviation of the topographic control of the ACC passing the Amsterdam–Kerguelen Island Passage causes a decrease in flow speed and subsequent deposition of previously eroded (local) material in the study area[14,65] (Supplementary Fig. 1). Second, vigorous AABW flow in the deep cyclonic gyre in the western AAB leads to erosion and formation of sediment-laden nepheloid layers in the deepest reaches of the western AAB, and subsequent release of these sediments in calmer and shallower flow regimes, for instance at our study site[14,65]. Small-grained sediments at our core site (i.e. smaller than foraminifera) were likely subject to transport through currents and redistribution processes, but likely originated from a local source around Kerguelen Plateau, and possibly Crozet Plateau[14].

**Radiocarbon measurements of planktic and benthic foraminifera.** Bulk wet sediment samples were freeze-dried, disintegrated in de-ionized water, washed over a 150-µm-sized sieve to remove fine-grained particles and subsequently oven-dried at 45 °C. The planktic foraminifer *N. pachyderma* and mixed benthic foraminifera (excluding *Pyrgo* spp. and agglutinated specimen) >150 µm were hand-picked from or near planktic foraminifer abundance maxima (Supplementary Fig. 11), and were ¹⁴C-dated with both the MICADAS Accelerator Mass Spectrometer (AMS)-200 kV at the University of Bern (gas- and graphite samples) and with the AMS Pelletron-2.6 MV system at the ARTEMIS ¹⁴C laboratory of the University of Paris-Saclay (graphite samples only). Although we sampled our study core at high resolution, age reversals outside analytical uncertainties and outliers are entirely absent. None of our foraminiferal ¹⁴C analyses were discarded.

For ¹⁴C analyses with the Bern-MICADAS, the benthic and planktic foraminiferal samples were size-matched and pretreated with a weak acid leach prior to complete acidification with orthophosphoric acid. The evolved sample $CO_2$ was then purified, mixed with He and directly injected into the gas ion source of the Bern-MICADAS, or for larger samples, transferred to the automated graphitization unit, subsequently pressed into targets and loaded into the target rack of the Bern-MICADAS[20]. We have measured duplicates (n = 12), triplicates (n = 1) and quadruplicates (n = 2), which indicates a ¹⁴C age reproducibility of gaseous samples within 200 ¹⁴C yr (n = 11) for planktic samples and 130 ¹⁴C yr (n = 4) for benthic samples throughout the deglaciation[20]. Mean sample sizes of our *N. pachyderma* and mixed benthic foraminiferal samples measured with the gas ion source, i.e. with the Bern-MICADAS, amount to 0.61 ± 0.20 mg CaCO₃ (n = 57) and 0.56 ± 0.20 mg CaCO₃ (n = 43), respectively. The mean sample size of our *N. pachyderma* samples measured as graphite with the Bern-MICADAS amounts to 0.84 ± 0.19 mg CaCO₃ (n = 7). Sample pretreatment and specifications of the ¹⁴C analyses with the Bern-MICADAS (including data correction, (foraminiferal) blanks, reproducibility and accuracy) are described in detail elsewhere[20]. For ¹⁴C analyses at the University of Paris-Saclay, monospecific samples of the planktic foraminifera *N. pachyderma*, *G. bulloides* and *G. inflata* were weakly acid-leached, dissolved with orthophosphoric acid and graphitized with hydrogen on iron powder at 600 °C prior to AMS analysis. Mean sample size of these planktic foraminifer samples amounts to 5.7 ± 4.2 mg CaCO₃ (n = 31).

**Bottom water oxygen reconstructions.** We have used three approaches to determine bottom water oxygenation changes at our core site over the last degla-ciation. The first two approaches provide qualitative oxygen reconstructions, and are based on the enrichment of redox-sensitive elements (such as U and Mn) in

bulk sediments and in authigenic coatings of foraminifera. The second proxy provides quantitative [O₂] estimates based on epibenthic to deep infaunal for-aminiferal δ¹³C gradients.

The (re)cycling of redox-sensitive elements such as Mn and U in marine sediments during early diagenesis is fueled by oxygen diffusion from bottom waters and the oxygen demand related to bacterial respiration of (labile) organic compounds. The precipitation of insoluble uranium compounds (i.e. uraninite) results from strong reducing conditions in pore waters, and occurs on any sedimentary grains such as detrital material and carbonate shells of foraminiferal tests, resulting in a co-variation of bulk sedimentary and foraminiferal coating aU levels[66]. The U/Ca ratio of bulk foraminifera is dominated by the U enrichment in foraminifer coatings, and was hence proposed as a qualitative indicator of bottom water [O₂] changes[6,66]. Foraminiferal U/Mn ratios may also reflect bottom water oxygenation changes similar to U/Ca ratios[6].

U/Ca- and U/Mn analyses were performed on oxidatively cleaned 10–60 specimens of the planktic foraminifer *N. pachyderma* (200–250 µm size) by inductively coupled plasma-mass spectrometry (ICP-MS)[67]. The reproducibility of replicate foraminiferal U/Ca and U/Mn analyses is within 30 nmol mol⁻¹ and 0.7 mmol mol⁻¹ (n = 13), respectively.

For bulk sedimentary U analyses, about 100 mg of dry and homogenized sediments were spiked with known quantities of ²²⁹Th, ²³³U and ²³⁶U and subsequently digested in a blend of concentrated nitric, hydrochloric and hydrofluoric acid in a pressure-assisted microwave (with maximum temperature set to 180 °C). Th and U fractions were then separated using anion exchange column chromatography (AG1-X8 resin), filtered and dissolved in 1 M HNO₃ prior to analysis by multi-collector (MC)-ICP-MS (Thermo Fisher Scientific Neptune Plus). The contribution of detrital ²³⁰Th has been estimated by assuming a ²³⁸U/²³²Th ratio of 0.6 ± 0.1 and a correction for the detrital ²³⁴U/²³⁸U not in secular equilibrium of 0.96 ± 0.04. The initial ²³⁴U/²³⁸U ratio is assumed to be that of seawater (1.147 ± 0.004). Standard errors are 3.6% and 1.2% for ²³⁸U and ²³⁴U, and 6.3% and 3.3% for ²³²Th and ²³⁰Th, respectively. These translate into an average uncertainty of estimated aU concentrations of 2.8% (2σ).

The δ¹³C difference between the deep infaunal benthic foraminifer *Globobulimina affinis*, which is considered to preferably live at the oxic-anoxic boundary, and the epifaunal/shallow infaunal benthic foraminifera *Cibicides kullenbergi* (Δδ¹³C) was proposed to be a measure of the pore water δ¹³C depletion during organic carbon respiration that is controlled by bottom water oxygen levels through downward diffusion into marine sediments[36,68]. We have performed stable isotopic analyses on 1–4 specimens of *G. affinis* and *C. kullenbergi* (both >150 µm) on Finnigan Δ+ and Elementar Isoprime mass spectrometers and apply the latest Δδ¹³C–[O₂] calibration[36] to quantify past bottom water [O₂] variations at our study site, which has a calibration uncertainty of 17 µmol kg⁻¹. The carbon isotopic composition is expressed as δ¹³C in ‰ versus Vienna Pee Dee Belemnite (VPDB). The mean external δ¹³C reproducibility of our carbonate standards is σ = 0.03‰, but the intra-species δ¹³C variability for both of our benthic foraminiferal species is much larger (*C. kullenbergi*: 0.28‰, n = 54; *G. affinis*: 0.19‰, n = 34).

We observe a significant offset of 0.30 ± 0.30‰ (n = 8) between core-top and Holocene *C. kullenbergi* and *C. wuellerstorfi* δ¹³C values as well as during deglacial and glacial periods (0.28 ± 0.30‰, n = 12) in sediment cores from the study area (MD12-3396CQ, MD12-3394, MD02-2488 and MD12-3401CQ; see locations in Supplementary Fig. 1). As *C. wuellerstorfi* is often considered a truly epibenthic-living foraminifera, the observed offset can be likely explained by a bias of *C. kullenbergi* δ¹³C towards more negative values because of a shallow infaunal habitat[56]. In order to correct for this potential habitat bias, we shift our *C. kullenbergi* δ¹³C record to more positive values by 0.30 ± 0.30‰ prior to calculation of Δδ¹³C and bottom water [O₂], and associated uncertainties.

**Chronology and ventilation age reconstructions.** In order to obtain a detailed chronology for our study core, we have estimated changes in planktic reservoir ages at the core site of MD12-3396CQ over the last 25 kyr. These estimates are based on ¹⁴C-independent age control points that were obtained by aligning (sub-)SST changes at MD12-3396CQ with changes in Antarctic temperature recorded in water isotope variations in the Antarctic ice-core EPICA (European Project for Ice Coring in Antarctica) Dome C (EDC)[25] applying the most recent AICC2012 ice chronology[69]. Deglacial and glacial (sub-)SST changes were estimated based on (i) planktic foraminiferal assemblages (mean calibration error: 1.1 °C), (ii) the lipid composition of marine archaeal biomarkers (TEX$^L_{86}$) using the TEX$^L_{86}$-tempera-ture calibration of ref. [23] (calibration error: 2.8 °C), and (iii) Mg/Ca ratios of the planktic foraminifera *N. pachyderma* (calibration error: 1.5 °C). To assess changes in planktic foraminiferal assemblages, at least 300 planktic specimens from a representative sample aliquot >150 µm were counted under the microscope and then calibrated with the Southern Ocean core-top dataset of ref. [21]. For TEX$^L_{86}$ determination, freeze-dried bulk sediment was extracted three times using a 1:1 mixture of dichloromethane and methanol in Accelerated Solvent Extractor (ASE 350) cells filled with 8 g of 5% deactivated silica (in hexane) following ref. [70]. The extracts were evaporated using a rocket solvent evaporator (Genevac-Thermo) and subsequently filtered using a PTFE filter (0.2 µm pore size) with a 1.8% mixture of hexane:isopropanol. Lipid biomarker were analysed using a high-performance liquid chromatograph (Agilent, 1260 Infinity) coupled to a single quadrupole mass

spectrometer detector (Agilent, 6130). Mg/Ca ratios were measured on 10–60 oxidatively cleaned specimens of *N. pachyderma* (200–250 μm fraction) via ICP-MS (ref. [67]). They were converted into SST using the calibration Mg/Ca = 0.406 × $e^{0.083 \times T}$ of ref. [22].

Our stratigraphic alignment of (sub-)SST variations at our core site to the EDC δD record is guided by the first principal component (PC1) of all three (sub-)SST datasets that explains 77% of the variance of all three records (Supplementary Figs. 2–4). Our approach makes the assumption of a fast (centennial-scale) thermal equilibration of circum-Antarctic surface waters and Antarctic air temperatures recorded at EDC through an atmospheric connection[5]. We assign an ad-hoc cal. age uncertainty of 500 yr to our age markers younger than 20 kyr BP, but use an uncertainty of 1000 yr for tiepoints from the core section older than 20 kyr BP owing to more subdued temperature variability for that period (Supplementary Figs. 2–4). For our tiepoint at 27.7 kyr, we consider an age uncertainty of 2000 yr, because the variability of PC1 and the reference Antarctic temperature record is low, and all three (sub-)SST records in our study core show strongest disagreement at that time (Supplementary Fig. 4).

We uncalibrated all age markers and converted them into $^{14}$C age-space based on the ShCal13 calibration[26]. This conversion considers both the uncertainties of the cal. age tiepoints and the atmospheric calibration, and was performed with the radcal software package[71] (Supplementary Fig. 5). Our high-resolution planktic $^{14}$C ages (and their $^{14}$C age uncertainties) are interpolated onto the depths of all (sub-) SST-based tiepoints. Using the now paired planktic foraminiferal and atmospheric $^{14}$C age constrains at our tiepoints, we compute the corresponding surface-ocean reservoir ages, i.e. the $^{14}$C age offsets between planktic foraminifera and the atmosphere at that time (i.e. d$^{14}$R$_{P-Atm}$; Supplementary Fig. 5). We report median d$^{14}$R$_{P-Atm}$ and the 66% (1σ) probability density range of each sample, as determined by the radcal.R script[71]. For the topmost sample (at 4.5 cm), we assume a prebomb surface-ocean reservoir age of d$^{14}$R$_{P-Atm}$ = 700 ± 150 yr (Supplementary Fig. 5; the uncertainty of 150 yr estimate incorporates possible centennial d$^{14}$R$_{P-Atm}$ variability in the study region[27]).

In order to subtract surface-ocean reservoir variations from our measured planktic $^{14}$C record, the d$^{14}$R$_{P-Atm}$ estimates were interpolated onto the depths where discrete planktic $^{14}$C measurements are available. Our 95 d$^{14}$R$_{P-Atm}$-corrected planktic $^{14}$C ages were then converted to cal. ages using the ShCal13 calibration[26] and the OxCal (version 4.3.2) program. The final age model of the upper core section (<426 cm) with dense age constraints is based on a Bayesian deposition model, obtained by a P_Sequence model within OxCal (P_Sequence ("MD12-3396CQ",0.07,0.2)) and an outlier assessment (Outlier_Model("General",T(5),U(0,4),"t")), which considers all calibrated planktic d$^{14}$R$_{P-Atm}$-corrected $^{14}$C ages and our tiepoints resulting from the stratigraphic alignment of (sub-)SST variations to Antarctic temperature (Supplementary Fig. 6). For the lower core section with sparser age constraints (>426 cm), the age-depth relationship was obtained through linear interpolation between our four lowermost (sub-)SST-based tiepoints and their ad-hoc age uncertainties (Supplementary Fig. 6). In this study, we focus, however, on the upper 400 cm of the core, which records the last deglacial and glacial periods with sedimentation rates ranging between 7–35 cm kyr$^{-1}$ (Supplementary Fig. 4).

We estimate changes in deep-ocean ventilation based on $^{14}$C age offsets between benthic and planktic foraminifera (d$^{14}$R$_{B-P}$) as well as between benthic foraminifera and the contemporaneous atmosphere (d$^{14}$R$_{B-Atm}$). We use the radcal.R script[71] to calculate d$^{14}$R$_{B-Atm}$, which considers 1σ-uncertainties of our chronology (i.e. our sediment deposition model), our measured $^{14}$C ages and the atmospheric calibration. We report the median and the 66%-probability density range of our deep-ocean ventilation age estimates as calculated by radcal.R[71].

At our study site, we find a non-negligible $^{14}$C background of $^{14}$C-free foraminifera, with a fraction modern F$^{14}$C = 0.0007–0.0009 for graphitized planktic foraminiferal samples, as well as F$^{14}$C = 0.0024–0.0031 for foraminifera measured in gaseous form[20]. The foraminiferal backgrounds were determined for the core site with the Bern-MICADAS, but similar estimates from the ARTEMIS facilities are lacking. Given the uncertainties associated with a consistent and reliable foraminiferal blank correction of our Bern-MICADAS and Paris-Saclay $^{14}$C data[20], we report all ventilation age reconstructions without consideration of foraminiferal $^{14}$C backgrounds. However, applying a foraminiferal background correction has a negligible impact on our d$^{14}$R$_{B-P}$ and d$^{14}$R$_{B-Atm}$ estimates during the last deglaciation, as it only slightly affects the absolute values of our results, mostly during the LGM, but not the observed trends (Supplementary Fig. 7).

**Transient model simulations**. Transient simulations of marine reservoir ages were performed with an enhanced version of the Hamburg LSG ocean general circulation model forced with temporal changes in atmospheric $^{14}$C climate under present-day (PD) and last glacial climate conditions (GS)[29]. Simulated surface- and deep-ocean reservoir ages in the study area (80–100°E, 45–50°S, 25 m and 3.5 km water depth, respectively) were extracted from previous model simulations[29]. The model results are compared against our proxy data to highlight dynamical ocean changes independent of transient changes in air-sea gas exchange, i.e. solely due to variations in Δ$^{14}$C$_{atm}$ and CO$_{2,atm}$ (grey lines in Fig. 4a, c)—two parameters that are used to transiently force the ocean general circulation model of ref. [29] under different climate conditions.

## Data availability

The datasets generated during the current study are available from the PANGAEA database (https://doi.org/10.1594/PANGAEA.912711). The Global Ocean Data Assimilation System (GODAS) database is available at https://psl.noaa.gov/data/gridded/data.godas.html. The data product of the Global Ocean Data Analysis Project version 2 can be found at https://www.ncei.noaa.gov/access/ocean-carbon-data-system/oceans/GLODAPv2_2019/. Atmospheric CO$_2$ concentrations are available in the supplement to https://doi.org/10.1002/2014GL061957. The ShCal13 calibration can be downloaded from http://www.radiocarbon.org/IntCal13%20files/shcal13.14c. The EPICA Dome C δD record is available at https://doi.org/10.1594/PANGAEA.683655, while the AICC2012 age scale is provided in the supplement to https://doi.org/10.5194/cp-9-1733-2013). The NGRIP δ$^{18}$O record is available at https://doi.org/10.1594/PANGAEA.586886 with the GICC05 age scale available at https://www.ncdc.noaa.gov/paleo/study/6086. Data of ref. [16] are available at https://doi.org/10.1594/PANGAEA.906365. Model simulation results of ref. [29] are available at https://doi.org/10.1594/PANGAEA.876733. Proxy data of MD07-3076CQ can be found under https://doi.org/10.1594/PANGAEA.845078, https://doi.org/10.1594/PANGAEA.861823 and https://science.sciencemag.org/content/suppl/2010/05/24/328.5982.1147.DC1. Data of ref. [55] are available in the supplement at https://doi.org/10.1016/j.palaeo.2010.08.011. Radiocarbon data of ref. [30] and refs. [31,33] are available at https://doi.org/10.1594/PANGAEA.823116 and in the supplement (https://doi.org/10.1016/j.epsl.2015.12.039) as well as through personal communication, respectively. Tasman coral data of ref. [35] are provided in the supplement at https://doi.org/10.1016/j.epsl.2015.09.038. Data of refs. [72,73] can be found in the associated supplement files (https://doi.org/10.1038/ncomms14203 and https://doi.org/10.17632/9v7ygwc9jj.1).

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

## Acknowledgements

J.G. and S.L.J. (grants PP00P2_144811 and 200021_163003), A.S.S. (grant PBEZP2_145695), and A.M.-G. (grant PZ00P2_142424) acknowledge funding from the Swiss National Science Foundation. J.G. was also supported by a Global Research Fellowship from the German Research Foundation (DFG grant GO 2294/2-1), and a promotion grant from the Intermediate Staff Association of the University of Bern. A.M. and E.M. acknowledge financial support from the French Ministry of Research and Higher Education, the Swedish Research Council (grant VR-349-2012-6278), and the French National Institute of Sciences of the Universe at the French National Centre for Scientific Research. A.M.-G. further acknowledges funding from the Max Planck Society. M.B. is supported by the German Federal Ministry of Education and Research (BMBF), a Research for Sustainability initiative (FONA, www.fona.de) through the PalMod project (grant number: 01LP1919A). We thank the French Polar Institute Paul-Émile Victor, the captains and the crew of RV *Marion Dufresne* during the Indien-Sud cruises for their help retrieving sediment core MD12-3396CQ. We also thank the staff of the Laboratoire de Mesure du Carbone-14 of the ARTEMIS French National AMS facility, the Laboratory for Radiocarbon Analysis at the University of Bern (especially Michael Battaglia and Gary Salazar), Christopher Bronk Ramsey, Gulay Isguder, Fabien Dewilde, Derek Vance and Corey Archer for technical support. We are indebted to Sophie Hines, Ning Zhao, Jerry McManus, Jimin Yu, Andy Hogg, Veronica Tamsitt, Spencer Jones, Danny Sigman, and Wally Broecker for insightful discussions.

## Author contributions

J.G., E.M., and S.L.J. devised the study. E.M. and A.M. collected the core material. J.G., E.M., L.M.T, A.S.S., A.P.H., N.S., M.B., A.M.G., and S.S. performed the analyses. J.G., E.M., A.S.S., A.P.H., and N.S. analyzed the proxy data. J.G. wrote the manuscript with contributions from all co-authors.

## Competing interests

The authors declare no competing interests.
