## [Peer Review File · Nature Communications]

Reviewers' Comments:

Reviewer #1:

Remarks to the Author:

A review (by Patrick A. Rafter) of the manuscript, "Fast deglacial carbon release from the South Indian Ocean 'upwelling hotspot' within centuries and millennia".

This manuscript provides an excellent collection of new proxy measurements and I entirely support its publication. I am listing a few, fairly important comments immediately below that I think will significantly improve the manuscript. These suggestions are then followed by line-by-line notes on what I liked, what I thought was confusing, and sometimes how I think the text could be improved.

First, I am interested to know why the Barker et al. (2010) deep South Atlantic site isn't discussed in more detail.

Second, there is A LOT (!) of good science in the Methods and Supplementary Figures. I think the manuscript will be improved if some of this examination of the current and previously published measurements was incorporated into the main body of the manuscript (I believe Nature Comms doesn't have page limits, so this should be ok). Among these hidden gems are sensitivity tests of both the Ronge (2020) and Skinner (2010) age models—both deserve much more detail than is provided here (perhaps as a new section in the Results?).

Third, I think the flow of the manuscript is good between the Introduction and the Results. (And I believe the flow will be improved with the addition of some of the Methods and Supplements.) But I found some of the text in the Discussion section to be repetitive and other text to be confusing. One source of my confusion (a typical reader of this manuscript) in the Discussion is that the text frequently moves backwards and forwards in time between different time periods. I think that, an ideal manuscript that would be well understood by most readers will move forward in time from the LGM to the Holocene. And the Discussion text (which is meant to bring together a holistic understanding of the new data in the context of published work) would ideally move across the time period of interest only once. The text would then have to carefully consider how to describe the *amazing new data* in a holistic way. The current manuscript describes one observation (14C), then another (oxygen proxies), then quantifies the carbon sequestration... this can be made into one coherent text. And this one coherent text to bind them all would probably be more concise. Otherwise, at a bare minimum, there should be headings in the Discussion that segregate it into different topics, but the text would still need to be adjusted so that it discusses each section moving forward in time through the relevant time period.

Line by line notes (as I read the manuscript):

Title: I would argue that the title doesn't fit the main subject of the manuscript. I would've named this manuscript something that highlights the major finding that there are spatial differences in Southern Ocean carbon storage. The centennial to millennial changes in carbon release (if you ask me) refers to a more speculative and less convincing aspect of the manuscript.

35: Isn't it more like 14.3-kyr BP? Or maybe "near the start of the B/A"?

41: I think "inventory" is better than "activity" but I guess either word is ok

71: Great sentence

103: I'm fine with this, but I imagine another reviewer might not like "robust age model" based on my personal experience...

107: I realize this is a major subject of this manuscript, but I would soften or qualify the statements of marked spatial differences between the deep Indian and Atlantic. For example, doesn't this statement entirely depend on assuming the original age model of the South Atlantic site? And my notes ask, where is the Barker et al. (2010) observations from the deep South Atlantic?

110: As I stated in my Summary above, I don't think it's useful to overly advertise the century to millennia changes—this dataset is *amazing*, but I don't think it is providing me (the typical reader of this work) much new information about changes in atmospheric CO₂ on these timescales.

112: I think it's ok to use former and latter sometimes, but this part of the manuscript (the short summary of what is coming up) is very important. I therefore think it's better to have a full, detailed sentence instead of requiring the reader to look back at the previous sentence to remind themselves what was "latter".

130: I think Mortyn et al. (2003) in *Geo*³ is a good reference for building age models tied to Antarctic air temps.

149-151: I would suggest rewording this. I found it confusing

160-164: This is confusing. The surface reservoir age estimates can barely be distinguished from pre-bomb values, but they are also good enough to say that the Ronge et al. (2020) surface reservoir ages are not good. I was very confused by this statement until I carefully read the supplementary figures and captions. This comparison (Fig. S10) is convincing! And I think it deserves much more explanation than what is provided in the current manuscript. I think it could probably be accomplished with a paragraph or two in the Results section, but not in the kind of confusing text in the Supplement.

229: does this mean lowered to a value of 100 or lowered from the present day by 100?

270-275: I like this.

278: My note in the margin of the manuscript says, "Why no Barker et al. (2010)?"

283: This is another sentence that was confusing until I looked at Fig. S8 and thought about it for some time. This is a very useful examination of the MD07-3076 age model and I am sure that other readers would also be interested in reading more about these sensitivity tests. Is it also worth considering a sensitivity test where the benthic $\delta^{18}O$ are matched? Seems that some of the MD07-3076 age model during the LGM could go younger, which would erase much of that difference between the South Indian and (this) South Atlantic site.

332: help explain the...

343: Isn't this late deglacial ventilation decrease more similar to ACR than the YD? This was also shown for the Ronge (2010) South Indian Ocean sites, which look even more similar to the new work in this manuscript with the adjusted age models!

352: My notes in the manuscript tell me that this shift in time periods is getting confusing.

363: This sentence can be more clear—what does the "this" refer to? And what exactly are the observations that detail a remarkably homogenous water column (I know what they are, but it would help the readers to see it stated clearly). Also, the Hines deep-sea coral ^{14}C ages are younger than the deeper sites during the ACR, so is it accurate to say there was vertical homogeneity.

381: I believe the benthic abundance variability was already mentioned on Line 328.

387-389: This sentence could be clarified. It's a little confusing.

390-392: Also could use clarification.

396: Could this text use some softening? It is a very strong statement, considering all the assumptions.

399: they point to

401: increases in South Indian convection as shown by...

The figures are all beautiful. The comparison of the upwelling zones across the Southern Ocean should be very informative. The map inset in Fig. 5 is so good that I'm going to steal that idea for upcoming papers.

The only comment I have is on Fig. 2. As far as I understand, there is no ^{14}C record within the South Australian Bight. Perhaps this is based on a mistake made in Skinner (2017), where the latitude for site KT89-18-P4 was mistakenly labeled as -32.2 N, when it is actually +32.2 N.

Reviewer #2:

Remarks to the Author:

Gottschalk et al, provide exciting new data documenting potential fast Southern Ocean carbon release during the last deglaciation, and highlight their study area as a potential 'upwelling hotspot.'

I think this is a very well written manuscript, supported by novel data. It is great to see that smaller volumes of samples material can now be analyzed to make these inferences. It is also good to see that the authors assess sample reproducibility, species offsets, etc. Importantly, the work shows that it is important to include assessments of the Indian Ocean with regards to atmospheric CO₂ variations during glacial-interglacial cycling.

I have one suggestion. Early on in the manuscript (lines 46 to 49) the authors indicate that ocean ¹⁴C is a transient tracer, and that ocean-versus-carbon ¹⁴C ultimately reflect the accumulation of respired carbon. It would be very interesting to see a graph illustrating this relationship today (something like ¹⁴C age versus AOU); the data should be easily available from NOAA.

Full point-by-point response to reviewers' comments on manuscript NCOMMS-20-19175

Fast deglacial carbon release from the South Indian Ocean ‘upwelling hotspot’ within centuries and millennia by Gottschalk et al.

We sincerely thank Patrick Rafter and the anonymous reviewer for the thorough evaluation of our manuscript, which we feel has substantially contributed to clarify our study. We have improved the manuscript in light of their comments, as described in detail below.

Both reviewers highlight the potential of our study in advancing our understanding of past changes in the marine carbon cycle dynamics on centennial- to orbital timescales in the past, providing novel insights from a hitherto understudied region, the South Indian Ocean. The reviewers also raised concerns, which we feel we can address adequately as outlined in our letter to the Editor and in our point-by-point response below.

Please note that line numbers below refer to the revised manuscript *with* tracked changes. Text in green below indicates revised text in our manuscript. Reference numbers in the document with and without tracked changes differ.

Reviewers' comments:

Reviewer #1 (Remarks to the Author):

A review (by Patrick A. Rafter) of the manuscript, “Fast deglacial carbon release from the South Indian Ocean ‘upwelling hotspot’ within centuries and millennia”.

This manuscript provides an excellent collection of new proxy measurements and I entirely support its publication. I am listing a few, fairly important comments immediately below that I think will significantly improve the manuscript. These suggestions are then followed by line-by-line notes on what I liked, what I thought was confusing, and sometimes how I think the text could be improved.

First, I am interested to know why the Barker et al. (2010) deep South Atlantic site isn’t discussed in more detail.

During the preparation of our manuscript, we debated whether we should include a detailed discussion of the *Barker et al.* [2010] data but finally decided not to for three main reasons. First, the *Barker et al.* [2010] record has poor surface ocean reservoir age control. *Barker et al.* [2010] used constant surface ocean reservoir ages, which according to the latest global compilation effort of *Skinner et al.* [2019] is inadequate. As this may lead to strong biases in deep-ocean ventilation age estimates, a direct comparison to our comprehensive ventilation age reconstructions (including direct estimates of surface ocean reservoir ages) would be flawed. Second, the TN057-21 record of *Barker et al.* [2010] is characterized by much lower data resolution (15 data points for the last 22 kyr BP) than our South Indian record (33 data points for the last 22 kyr BP) and South Atlantic site MD07-3076CQ used for comparison (30 data

points for last 22 kyr BP). In fact, the record of *Barker et al.* [2010] does not allow for a thorough analysis of centennial-scale ventilation age changes, and millennial-scale variations might be obscured by the assumption of constant surface ocean reservoir age changes. Thirdly, the hydrography at our South Indian study site (water depth 3.6 km, bathed in Lower Circumpolar Deep Water) more closely resembles the site described in *Skinner et al.* [2010] (MD07-3076Q: water depth 3.8 km, bathed in Lower Circumpolar Deep Water) when compared to the study considered in *Barker et al.* [2010] (TN057-21: water depth 4.9 km), which is under much stronger influence of Antarctic Bottom Water in the semi-enclosed, deep Cape Basin.

A robust comparison between our new South Indian data and the *Barker et al.* [2010] data would require a reevaluation of Barker's age model, the reconstruction of surface ocean reservoir ages at the site and ideally, an increase in data resolution. We feel that this effort is beyond the scope of our manuscript and beyond the length requirement for our manuscript, and would ideally require a separate study. Nonetheless, we argue that given similar methodology, data resolution and hydrographic setting, the South Atlantic core MD07-3076Q [*Skinner et al.*, 2010] is much better suited for comparison with our South Indian study site than the *Barker et al.* [2010] record. We now emphasize this in the main text: "In contrast, upstream in the South Atlantic at 3.8 km water depth⁵ (a site chosen because of a comparable hydrography and methodological approach used for our study core), glacial [...]" (line 216-217).

Second, there is A LOT (!) of good science in the Methods and Supplementary Figures. I think the manuscript will be improved if some of this examination of the current and previously published measurements was incorporated into the main body of the manuscript (I believe Nature Comms doesn't have page limits, so this should be ok). Among these hidden gems are sensitivity tests of both the Ronge (2020) and Skinner (2010) age models—both deserve much more detail than is provided here (perhaps as a new section in the Results?).

In our manuscript, we have tried to explore every possible explanation and made several sensitivity tests to strengthen our argumentation, for instance related to the differences in glacial ventilation ages between the South Atlantic and Indo-Pacific by critically assessing the MD07-3076CQ age model (Supplementary Fig. S8), including additional proxy data (Supplementary Fig. S9), or testing the consistency of our results with those of *Ronge et al.* [2020], when similar age model approaches are adopted (Supplementary Fig. S10). As our study is already quite extensive given the multi-proxy approach and high-resolution analyses of both of our ¹⁴C and [O₂] proxies, we were forced to move these sensitivity tests to the supplement in order to abide by *Nature Communications* formatting requirements. Moreover, although critical to ascertain the robustness of our argumentation, we feel that these aspects of the discussion are quite technical.

However, we understand Reviewer 1's comment, and explored the possibility of moving parts of the supplement to the main text. We feel that moving the sensitivity test of the *Skinner et al.* [2010] age model to the main text would inflate the discussion excessively, and would take the focus away from the new data. Additionally, similar sensitivity tests have previously been performed by *Burke and Robinson* [2012], which we now refer to in the main text. Because of these reasons, we think that the *Skinner et al.* [2010] sensitivity test is best placed in the supplement. However, we have expanded the main text and provide the reader with more information on these aspects: "Second, glacial $d^{14}R_{P-Atm}$ estimates in South Atlantic core MD07-3076CQ reach values larger than 2000 yr during the LGM [*Skinner et al.*, 2010], which may be unrealistic according to new compilations [*Skinner et al.*, 2019]. Nonetheless, disregarding these extreme $d^{14}R_{P-Atm}$ values (following similar sensitivity tests made by ref. [*Burke and Robinson*, 2012]) reduces but does not eradicate the observed LGM $d^{14}R_{B-Atm}$ mismatch with our South Indian study site (Supplementary Fig. S8)." (line 306-311).

[REDACTED]

et al. [2020] ventilation age reconstructions, simply because we can only test the effect of changing surface ocean reservoir ages on their records but cannot change the entire age model (data not available). As Supplementary Fig. 10 shows, the effect is important, and brings their data in alignment with our findings (although one has to emphasize the lower resolution of the *Ronge* datasets during the glacial and deglaciation as a caveat). We have provided the reader with some additional information in the main text to emphasize this finding “We show that the disagreement results from insufficiently accounted surface ocean reservoir age variability **that can be improved by applying consistent surface ocean reservoir ages for all sites** (Supplementary Fig. S10).” (line 449-452). However, we acknowledge that given the *Nature Communications* formatting requirements [REDACTED] we are inclined *not* to move the sensitivity test discussion to the main text. We hope that our reasons for not moving the *Ronge et al.* [2020] sensitivity to the main text are justified.

Third, I think the flow of the manuscript is good between the Introduction and the Results. (And I believe the flow will be improved with the addition of some of the Methods and Supplements.) But I found some of the text in the Discussion section to be repetitive and other text to be confusing. One source of my confusion (a typical reader of this manuscript) in the Discussion is that the text frequently moves backwards and forwards in time between different time periods. I think that, an ideal manuscript that would be well understood by most readers will move forward in time from the LGM to the Holocene. And the Discussion text (which is meant to bring together a holistic understanding of the new data in the context of published work) would ideally move across the time period of interest only once. The text would then have to carefully consider how to describe the *amazing new data* in a holistic way. The current manuscript describes one observation (14C), then another (oxygen proxies), then quantifies the carbon sequestration... this can be made into one coherent text. And this one coherent text to bind them all would probably be more concise. Otherwise, at a bare minimum, there should be headings in the Discussion that segregate it into different topics, but the text would still need to be adjusted so that it discusses each section moving forward in time through the relevant time period.

We agree with Reviewer 1 that ideally the discussion should proceed chronologically from the LGM to the Holocene. We have attempted to follow exactly this strategy in our original manuscript, by beginning with a discussion centered on the LGM, followed by HS1 and the BA. As we find that processes during the YD and the onset of the Holocene were akin to those at play during HS1 and the onset of the BA warm period, respectively, we have addressed these intervals while discussing the HS1 and the BA onset. We suspect that this combination of a chronological discussion of events and a process-oriented discussion may have caused confusion. The comments brought up by Reviewer 1 hence prompted us to revise this approach.

We have hence restructured the discussion to follow a chronological progression in discussing the most important events of our proxy records (lines 371-457).

The inclusion of sub-headings in the discussion section is unfortunately not allowed according to *Nature Communications* formatting guidelines. Nonetheless, we hope that our revised discussion following a more straightforward chronological description will improve the flow and clarity of our argumentation.

Line by line notes (as I read the manuscript):

Title: I would argue that the title doesn't fit the main subject of the manuscript. I would've named this manuscript something that highlights the major finding that there are spatial differences in Southern Ocean carbon storage. The centennial to millennial changes in carbon release (if you ask me) refers to a more speculative and less convincing aspect of the manuscript.

We feel that it is important that the title reflects the major scientific highlights of the manuscript. These relate in our view to *both* the observed regional heterogeneity in carbon storage in the glacial Southern Ocean and the distinct, yet hitherto unrecognized, role of the South Indian Ocean in releasing carbon on centennial- and millennial timescales. Because we are able to compare our proxy data with similar high-resolution data from the South Atlantic and south of Tasmania (all sites with comprehensive age control), we consider our interpretations of centennial- and millennial-scale ^{14}C and O_2 ventilation changes in our study region as relevant and robust.

Acknowledging Reviewer 1's recommendation, we came up with an alternative title that we hope better reflects the novelty and important findings of our study: *Glacial heterogeneity in Southern Ocean carbon storage abated by fast South Indian deglacial carbon release*

35: Isn't it more like 14.3-kyr BP? Or maybe "near the start of the B/A"?

Reviewer 1 is correct in prompting us to clarify this timing. We have clarified that "The dissipation of this heterogeneity commenced 14.6 kyr ago." (line 37).

41: I think "inventory" is better than "activity" but I guess either word is ok

Amended.

71: Great sentence

Thank you.

103: I'm fine with this, but I imagine another reviewer might not like “robust age model approach” based on my personal experience...

We replaced “robust age model approach” with “comprehensive age model approach”.

107: I realize this is a major subject of this manuscript, but I would soften or qualify the statements of marked spatial differences between the deep Indian and Atlantic. For example, doesn't this statement entirely depend on assuming the original age model of the South Atlantic site? And my notes ask, where is the Barker et al. (2010) observations from the deep South Atlantic?

We agree that this is a major finding of our study, and hence took great care to incorporate additional proxy evidence in our original manuscript in support of the observed glacial South Atlantic versus South Indian ^{14}C ventilation age differences (both sites provide high-resolution proxy-data and follow an identical age model approach, and are hence ideally suited for a direct comparison). Additional proxy support includes marked differences in $\Delta\delta^{13}\text{C}$ -based $[\text{O}_2]$ estimates and benthic $\delta^{13}\text{C}$ and $\delta^{18}\text{O}$ values at these two sites, as shown in Supplementary Fig. S9. The observed differences in these three geochemical parameters remain robust, even when reasonable changes to the age model of the South Atlantic site are considered (Supplementary Fig. S8, S9). In fact, we show that reasonable adjustments to the age model of the South Atlantic site does not eliminate the observed inter-basin offsets in ^{14}C ventilation (Supplementary Fig. S8). We hence believe that the inferred spatial heterogeneity in Southern Ocean carbon storage is an important and novel insight of our study. We revised the text to emphasize the multi-proxy support of our statement: “We show based on multi-proxy evidence that, while the deep South Indian Ocean was a significant (remineralized) carbon sink during the last glacial, marked glacial inter-basin differences in carbon storage existed [...]” (line 112).

We have outlined the reasons that led us not to discuss the data presented by *Barker et al.* [2010] (see page 1 above). A robust comparison between our study site and that of *Barker et al.* [2010] requires an estimation of surface ocean reservoir ages, a reassessment of the age model and an increase in benthic ^{14}C data for the Barker site, which goes beyond the scope of our study.

110: As I stated in my Summary above, I don't think it's useful to overly advertise the century to millennia changes—this dataset is *amazing*, but I don't think it is providing me (the typical reader of this work) much new information about changes in atmospheric CO_2 on these timescales.

We have revised the text to focus more on processes rather than timescales. It now reads: “The dissipation of these regional glacial differences was mediated by a reinvigoration of Southern Ocean mixing during the first half of HS1 and enhanced Atlantic overturning at the onset of the BA interstadial, respectively, which we argue promoted a rise in $\text{CO}_{2,\text{atm}}$ levels.” (line 116-120).

We believe that our millennial- and centennial-scale proxy records are important and provide novel insights into marine carbon cycling in the South Indian Ocean during the last deglaciation – a region that has hitherto been poorly documented. First, we are able to show for the first time that millennial-scale changes in deep Southern Ocean (both ^{14}C and O_2) ventilation existed in the South Indian during early HS1 only and likely impacted atmospheric CO_2 changes between ~18.3 and ~16.3 kyr BP. Second, the onset of the BA interstadial coincided with a rapid “flushing” at our South Indian study area that is identical in timing and magnitude to similar observations in the Equatorial- [Chen *et al.*, 2015] and South Atlantic [Skinner *et al.*, 2010], which we find remarkable. To our knowledge, such a vast spatial expression of a “flushing event” was not shown before. We are convinced that our findings shed new light on the ocean’s role in driving fast atmospheric CO_2 change.

We have revised the main text of the discussion to convey the novelty of these findings in a clearer way: “A reduction in carbon release from the South Indian to the atmosphere during late HS1, likely due to an unfavorable superposition of the South Indian water column density structure with bathymetry around the Kerguelen Island, may have halted the early deglacial $\text{CO}_{2,\text{atm}}$ rise and promoted the plateauing of $\text{CO}_{2,\text{atm}}$ between ~16.3 and ~14.8 kyr BP (Fig. 4) [Marcott *et al.*, 2014]. The fact that this late HS1 return to glacial conditions in ^{14}C and O_2 ventilation is not observed in the South Atlantic [Skinner *et al.*, 2010; Gottschalk *et al.*, 2016] or elsewhere in the Southern Ocean indicates that inter-basin differences persisted throughout HS1, attributing the South Indian Ocean a unique role in driving deglacial $\text{CO}_{2,\text{atm}}$ variability.” (line 384-391). And “This may have caused a “flushing” of the southern, high-latitude carbon pool that is not only limited to the equatorial Atlantic⁶⁰ but expanded into the South Atlantic⁵ and South Indian Ocean (this study), with remarkable near-identical ventilation changes in the latter two regions and a much wider spatial impact on the global ocean that previously assumed (Fig. 5a).” (line 405-409).

112: I think it’s ok to use former and latter sometimes, but this part of the manuscript (the short summary of what is coming up) is very important. I therefore think it’s better to have a full, detailed sentence instead of requiring the reader to look back at the previous sentence to remind themselves what was “latter”.

We replaced “the latter” with “We find that increased Atlantic overturning at the start of the BA period [...]” (line 120).

130: I think Mortyn *et al.* (2003) in *Geo*³ is a good reference for building age models tied to Antarctic air temps.

Reviewer 1 is correct in referring to the study of *Mortyn et al.* [2003] as one of the first studies that established age models based on a stratigraphic alignment to Antarctic air temperature. *Mortyn et al.* [2003] used *G. bulloides* $\delta^{18}\text{O}$ as approximation of sea surface temperature, whereby in our study we reconstruct sea (sub-)surface temperature via three independent proxy approaches. Rather than citing *Mortyn et al.* [2003], given the mentioned methodological differences, we refer to one of the many earlier studies that has followed an approach similar to ours, namely *Skinner et al.* [2010]. We refrain from adding more appropriate citations (e.g., [Govin *et al.*, 2009; Vázquez Riveiros *et al.*, 2013; Waelbroeck *et al.*, 2019]) given *Nature Communications* formatting guidelines.

149-151: I would suggest rewording this. I found it confusing

We have reworded this confusing statement: “We find that past $d^{14}\text{R}_{\text{P-Atm}}$ ages deviate from pre-industrial (i.e., pre-bomb) surface ocean reservoir ages of 700 ± 150 yr at our study site (Fig. 4, Supplementary Fig. S5)^{28,29}” (line 160-162).

160-164: This is confusing. The surface reservoir age estimates can barely be distinguished from pre-bomb values, but they are also good enough to say that the Ronge *et al.* (2020) surface reservoir ages are not good. I was very confused by this statement until I carefully read the supplementary figures and captions. This comparison (Fig. S10) is convincing! And I think it deserves much more explanation than what is provided in the current manuscript. I think it could probably be accomplished with a paragraph or two in the Results section, but not in the kind of confusing text in the Supplement.

[REDACTED] As outlined above, we have added some additional information to the text, in order to improve the clarity of our argumentation of this context, while at the same time abiding by *Nature Communications* formatting requirements. Based on Supplementary Fig. S10 we show that consistent surface ocean reservoir ages for the study region will bring both studies (ours and *Ronge et al.* [2020]) in accordance with each other, which is ultimately the most important point that we want to make. In our view, this does not require a rather technical description of the details of the *Ronge et al.* [2020] study in the discussion.

229: does this mean lowered to a value of 100 or lowered from the present day by 100?

We have corrected this statement following the reviewer’s recommendation stating that “bottom water $[\text{O}_2]$ during the LGM was lowered by 100 ± 40 $\mu\text{mol kg}^{-1}$ from present-day concentrations” (line 250).

270-275: I like this.

278: My note in the margin of the manuscript says, “Why no Barker et al. (2010)?”

See our response to this important comment above (page 1).

283: This is another sentence that was confusing until I looked at Fig. S8 and thought about it for some time. This is a very useful examination of the MD07-3076 age model and I am sure that other readers would also be interested in reading more about these sensitivity tests. Is it also worth considering a sensitivity test where the benthic $\delta^{18}\text{O}$ are matched? Seems that some of the MD07-3076 age model during the LGM could go younger, which would erase much of that difference between the South Indian and (this) South Atlantic site.

Any age model is complex and requires in-depth discussion. *Skinner et al.* [2010] has presented a very comprehensive age model and sediment deposition model for core MD07-3076CQ that was arguably the best available “solution” in the Southern Ocean at the time (given comprehensive estimates of surface ocean reservoir ages and high-resolution ^{14}C dates). Reviewer 1 is correct in stating that sensitivity studies regarding the robustness of the age model of core MD07-3076Q (or in fact any other core) would be valuable information for the community. *Burke and Robinson* [2012] have performed such useful sensitivity tests of the MD07-3076Q age model already, which we now refer to in our study. We also expanded the main text of the discussion to provide the reader with more information on our sensitivity test, while avoiding inflating the text and disturbing the chronological flow of the discussion of our new data: “Second, glacial $\delta^{14}\text{R}_{\text{P-Atm}}$ estimates in South Atlantic core MD07-3076CQ reach values larger than 2000 yr during the LGM [*Skinner et al.*, 2010], which may be unrealistic according to new compilations [*Skinner et al.*, 2019]. Nonetheless, disregarding these extreme $\delta^{14}\text{R}_{\text{P-Atm}}$ values (following similar sensitivity tests made by ref. [*Burke and Robinson*, 2012]) reduces but does not eradicate the observed LGM $\delta^{14}\text{R}_{\text{B-Atm}}$ mismatch with our South Indian study site (Supplementary Fig. S8).” (line 306-311). We feel, however, that given the focus of our paper on the deep South Indian Ocean and the word limitations for the main text as per *Nature Communications* guidelines that the sensitivity study of the MD07-3076Q age model is best placed in the supplement. We hope that our additional explanation and reference to *Burke and Robinson* [2012] in the main text alleviate some of the valid concerns of Reviewer 1.

As shown in Supplementary Fig. S9, the benthic $\delta^{18}\text{O}$ signatures of the two cores, MD07-3076Q and MD12-3396Q agree well during the last deglaciation. We consider the offsets in benthic $\delta^{18}\text{O}$ during the LGM to be real rather than an artifact of either or both age models, because a 4 kyr-shift of either of these age models during the LGM (to achieve agreement between the

benthic $\delta^{18}\text{O}$ signatures) would cause unrealistic changes in the sedimentation rates of these cores.

332: help explain the...

Amended.

343: Isn't this late deglacial ventilation decrease more similar to ACR than the YD? This was also shown for the Ronge (2010) South Indian Ocean sites, which look even more similar to the new work in this manuscript with the adjusted age models!

Figure 4 and 5 demonstrate a striking difference in absolute ^{14}C ventilation ages and aU-derived oxygenation during the ACR and YD at our South Indian study site. Decreased ventilation during the late deglaciation that we refer to in this text paragraph occurs after the end of the ACR, and hence parallels the YD stadial. With more reasonable and consistent surface ocean reservoir ages, the *Ronge et al.* [2020] data are entirely in support of this finding (Supplementary Fig. S10). We have clarified this statement accordingly, as we suspect that our original phrasing has caused some unintended confusion. The revised statement reads: “During the YD, we observe decreased ^{14}C and O_2 ventilation at our deep South Indian study site, suggesting [...]” (line 444-445). And “At the end of the YD, we observe a convergence of ventilation characteristics for different parts of the Southern Ocean, reminiscent of the 14.6 kyr BP-event (Fig. 5, Supplementary Fig. S9).” (line 452-453).

352: My notes in the manuscript tell me that this shift in time periods is getting confusing.

We have streamlined the discussion section, as described in detail in response to a previous comment above (please see page 4).

363: This sentence can be more clear—what does the “this” refer to? And what exactly are the observations that detail a remarkably homogenous water column (I know what they are, but it would help the readers to see it stated clearly). Also, the Hines deep-sea coral ^{14}C ages are younger than the deeper sites during the ACR, so is it accurate to say there was vertical homogeneity.

We have rephrased these sentences for clarification and specified the observations that led to our statement. These observations are highlighted in Fig. 5. The revised sentences now read “Because our South Indian ventilation age reconstructions closely agree with similar data from the deep South Atlantic [*Skinner et al.*, 2010], the Southwest Pacific [*Skinner et al.*, 2015; *Sikes et al.*, 2016] and intermediate water depths south of Tasmania [*Hines et al.*, 2015] within uncertainties (Fig. 5, Supplementary Fig. S9), we argue that reinvigorated deep-ocean ventilation led to [...]” (line 434-437).

Also, we respectfully disagree with the observation made by Reviewer 1 that the intermediate coral ^{14}C ages are younger than the deep sites during the ACR. As we show in Fig. 5, a clear distinction between our South Indian data (blue) and the intermediate-depth data (purple, *Hines et al.* [2015]) cannot be made given the uncertainties inherent to each dataset (both datasets in fact overlap with each other). We hence point out that our statement is valid within the confines of the given uncertainties of the datasets, and hope that this suffices to address the valid comment of Reviewer 1.

381: I believe the benthic abundance variability was already mentioned on Line 328.

Both instances are linked to two different time periods, namely during the early HS1 and the onset of the BA – both characterized by increased ^{14}C ventilation. This temporal distinction is important, and we have hence modified the main text to emphasize the observed benthic abundance variability during these different time periods.

387-389: This sentence could be clarified. It's a little confusing.

We have clarified this sentence as recommended: “A new equilibrium in air-sea gas exchange, however, was reached during the subsequent ACR period, when vertical mixing in the southern, high latitudes was seemingly stronger than at present-day as shown by lower-than-pre-bomb ventilation ages in the deep South Atlantic [*Skinner et al.*, 2010], South Indian (this study) and Southwest Pacific [*Sikes et al.*, 2016] (Fig. 5a).” (lines 431-434).

390-392: Also could use clarification.

We have revised this sentence: “Given an expansion of Antarctic sea ice cover [*Rae et al.*, 2018] and northern-hemisphere warming reducing the ocean CO_2 solubility [*Bauska et al.*, 2016] during the ACR, the capacity of the South Indian Ocean to impact $\text{CO}_{2,\text{atm}}$ levels during this time was likely limited despite high mixing rates commencing at ~14.6 kyr BP, which is consistent with the observed $\text{CO}_{2,\text{atm}}$ plateau during the ACR (Fig. 5a, b).” (lines 438-442).

396: Could this text use some softening? It is a very strong statement, considering all the assumptions.

We have toned down our statement while emphasizing the striking role of South Indian marine carbon cycle dynamics in the deglacial global carbon cycle. We changed the statement of our initial version of the manuscript “Overall, we identify unique fingerprints of marine carbon cycling in the South Indian Ocean on glacial and deglacial $\text{CO}_{2,\text{atm}}$ variations, mediated through forcing mechanisms in the northern and southern hemispheres.” to “Based on our high-resolution multi-proxy analyses, we identify marked impacts of marine carbon cycling in the South Indian Ocean on glacial and deglacial $\text{CO}_{2,\text{atm}}$ variations.” (line 459-460). Thereby, we acknowledge that the processes observed in the South Indian Ocean are not *per se* unique but

strikingly different in magnitude and timing that warrant consideration of a role of the South Indian Ocean in atmospheric CO₂ dynamics during the last deglaciation. This has not been comprehensively shown previously. We have removed the more speculative part on the driving mechanisms, and elaborate on them in more detail in the main text further below.

399: they point to

Amended.

401: increases in South Indian convection as shown by...

We have adopted this and added “as shown by both ¹⁴C (d¹⁴R_{B-Atm}) and O₂ ventilation proxies (aU, δΔ¹³C and foraminiferal U/Mn)” (line 467).

The figures are all beautiful. The comparison of the upwelling zones across the Southern Ocean is very informative. The map inset in Fig. 5 is so good that I'm going to steal that idea for upcoming papers. The only comment I have is on Fig. 2. As far as I understand, there is no ¹⁴C record within the South Australian Bight. Perhaps this is based on a mistake made in Skinner (2017), where the latitude for site KT89-18-P4 was mistakenly labeled as -32.2 N, when it is actually +32.2 N.

Thanks to the in-depth knowledge of Reviewer 1, we have corrected this mistake and have removed this incorrect core location from Figure 2.

Reviewer #2:

Gottschalk et al, provide exciting new data documenting potential fast Southern Ocean carbon release during the last deglaciation, and highlight their study area as a potential 'upwelling hotspot'.

I think this is a very well written manuscript, supported by novel data. It is great to see that smaller volumes of samples material can now be analyzed to make these inferences. It is also good to see that the authors assess sample reproducibility, species offsets, etc. Importantly, the work shows that it is important to include assessments of the Indian Ocean with regards to atmospheric CO₂ variations during glacial-interglacial cycling.

I have one suggestion. Early on in the manuscript (lines 46 to 49) the authors indicate that ocean ¹⁴C is a transient tracer, and that ocean-versus-carbon ¹⁴C ultimately reflect the accumulation of respired carbon. It would be very interesting to see a graph illustrating this relationship today (something like ¹⁴C age versus AOU); the data should be easily available from NOAA.

The close relationship between ¹⁴C levels and respired carbon content of seawater in the global ocean has previously been emphasized multiple times based on comprehensive datasets (e.g. NOAA or GLODAP). An extensive demonstration of this link was recently proposed by

Sarnthein et al. [2013] – the study we cite in the introduction in line 53. In our study, we follow the lead of *Sarnthein et al.* [2013] demonstrating in Fig. 7 the relationship between seawater ^{14}C age and $[\text{O}_2]$, the latter a direct function of AOU and respired carbon levels of seawater, using the newest version of the Global Ocean Data Analysis Project version 2 [*Olsen et al.*, 2016]. In addition, we make the first attempt to map this relationship in the past across ocean basins, taking advantage of combined reconstructions of seawater ^{14}C and $[\text{O}_2]$ – one of the few sites where this is possible includes our new data presented in this study. We hence believe that both our citation and Fig. 7 suffice to address the above comment of Reviewer 2.

Editorial changes

We have added a “Data Availability” section after the Methods section, and specify that “The datasets generated during the current study are available from the PANGAEA database (<https://doi.pangaea.de/10.1594/PANGAEA.912711>).”. A section on “Code Availability” is not warranted.

References

- Barker, S., G. Knorr, M. J. Vautravers, P. Diz, and L. C. Skinner (2010), Extreme deepening of the Atlantic overturning circulation during deglaciation, *Nat. Geosci.*, 3(8), 567–571, doi:10.1038/NGEO921.
- Bauska, T. K., D. Baggenstos, E. J. Brook, A. C. Mix, S. A. Marcott, V. V. Petrenko, H. Schaefer, J. P. Severinghaus, and J. E. Lee (2016), Carbon isotopes characterize rapid changes in atmospheric carbon dioxide during the last deglaciation, *Proc. Natl. Acad. Sci.*, 113(13), 3465–3470, doi:10.1073/pnas.1513868113.
- Burke, A., and L. F. Robinson (2012), The Southern Ocean’s Role in Carbon Exchange During the Last Deglaciation, *Science*, 335(6068), 557–561, doi:10.1126/science.1208163.
- Butzin, M., P. Köhler, and G. Lohmann (2017), Marine radiocarbon reservoir age simulations for the past 50,000 years, *Geophys. Res. Lett.*, 44, 8473–8480, doi:10.1002/2017GL074688.
- Chen, T., L. F. Robinson, A. Burke, J. R. Southon, P. Spooner, P. J. Morris, and H. C. Ng (2015), Synchronous Sub-millennial Scale Abrupt Events in the Ocean and Atmosphere during the Last Deglaciation, *Science*, 349(6255), 1537–1541, doi:10.1126/science.aac6159.
- Gottschalk, J., L. C. Skinner, J. Lippold, H. Vogel, N. Frank, S. L. Jaccard, and C. Waelbroeck (2016), Biological and physical controls in the Southern Ocean on past millennial-scale atmospheric CO_2 changes, *Nat. Commun.*, 7(11539), 1–11, doi:10.1038/ncomms11539.
- Govin, A., E. Michel, L. Labeyrie, C. Waelbroeck, F. Dewilde, and E. Jansen (2009), Evidence for northward expansion of Antarctic Bottom Water mass in the Southern Ocean during the last glacial inception, *Paleoceanography*, 24(26), 1202, doi:10.1029/2008PA001603.
- Hines, S. K. V., J. R. Southon, and J. F. Adkins (2015), A high-resolution record of Southern Ocean intermediate water radiocarbon over the past 30,000 years, *Earth Planet. Sci. Lett.*, 432, 46–58, doi:10.1016/j.epsl.2015.09.038.
- Marcott, S. A. et al. (2014), Centennial-scale changes in the global carbon cycle during the last deglaciation,

- Nature*, 514(7524), 616–619, doi:10.1038/nature13799.
- Mortyn, P. G., C. D. Charles, U. S. Ninnemann, K. Ludwig, and D. A. Hodell (2003), Deep sea sedimentary analogs for the Vostok ice core, *Geochemistry Geophys. Geosystems*, 4, 21, doi:10.1029/2002gc000475.
- Olsen, A. et al. (2016), The Global Ocean Data Analysis Project version 2 (GLODAPv2) – an internally consistent data product for the world ocean, *Earth Syst. Sci. Data*, 8, 297–323, doi:10.5194/essd-8-297-2016.
- Pérez-Tribouillier, H., T. L. Noble, A. T. Townsend, A. R. Bowie, and Z. Chase (2020), Quantifying lithogenic inputs to the North Southern Ocean using the long-lived thorium isotopes, *Front. Mar. Sci.*, 7(207), 1–16, doi:10.3389/fmars.2020.00207.
- Rae, J. W. B. et al. (2018), CO₂ storage and release in the deep Southern Ocean on millennial to centennial timescales, *Nature*, 562, 569–573, doi:10.1038/s41586-018-0614-0.
- Ronge, T. A., M. Prange, G. Mollenhauer, M. Ellinghausen, G. Kuhn, and R. Tiedemann (2020), Radiocarbon Evidence for the Contribution of the Southern Indian Ocean to the Evolution of Atmospheric CO₂ Over the Last 32,000 Years, *Paleoceanogr. Paleoclimatology*, 35(3), 1–16, doi:10.1029/2019pa003733.
- Sarnthein, M., B. Schneider, and P. M. Grootes (2013), Peak glacial ¹⁴C ventilation ages suggest major draw-down of carbon into the abyssal ocean, *Clim. Past*, 9(6), 2595–2614, doi:10.5194/cp-9-2595-2013.
- Sikes, E. L., M. S. Cook, and T. P. Guilderson (2016), Reduced deep ocean ventilation in the Southern Pacific Ocean during the last glaciation persisted into the deglaciation, *Earth Planet. Sci. Lett.*, 438, 130–138, doi:10.1016/j.epsl.2015.12.039.
- Skinner, L. C., S. Fallon, C. Waelbroeck, E. Michel, and S. Barker (2010), Ventilation of the deep Southern Ocean and deglacial CO₂ rise, *Science*, 328(5982), 1147–1151, doi:10.1126/science.1183627.
- Skinner, L. C., I. N. McCave, L. Carter, S. Fallon, A. E. Scrivner, and F. Primeau (2015), Reduced ventilation and enhanced magnitude of the deep Pacific carbon pool during the last glacial period, *Earth Planet. Sci. Lett.*, 411, 45–52, doi:10.1016/j.epsl.2014.11.024.
- Skinner, L. C., F. Muschitiello, and A. E. Scrivner (2019), Marine reservoir age variability over the last deglaciation: implications for marine carbon cycling and prospects for regional radiocarbon calibrations, *Paleoceanogr. Paleoclimatology*, 34, 1807–1815, doi:10.1029/2019PA003667.
- Thöle, L. M. et al. (2019), Glacial-interglacial dust and export production records from the Southern Indian Ocean, *Earth Planet. Sci. Lett.*, 525, 115716, doi:10.1016/j.epsl.2019.115716.
- Vázquez Riveiros, N., C. Waelbroeck, L. C. Skinner, J. C. Duplessy, J. F. McManus, E. S. Kandiano, and H. A. Bauch (2013), The “MIS 11 paradox” and ocean circulation: Role of millennial scale events, *Earth Planet. Sci. Lett.*, 371–372, 258–268, doi:10.1016/j.epsl.2013.03.036.
- Waelbroeck, C. et al. (2019), Consistently dated Atlantic sediment cores over the last 40 thousand years, *Sci. Data*, 6(165), 1–12, doi:10.1038/s41597-019-0173-8.

Reviewers' Comments:

Reviewer #1:

Remarks to the Author:

The authors have completely addressed my comments / suggestions and the final manuscript will be an excellent and useful addition to the scientific literature.

Reviewer #2:

Remarks to the Author:

I am happy with the response of the authors to my query, and look forward to seeing the work published.

Glacial heterogeneity in Southern Ocean carbon storage abated by fast South Indian deglacial carbon release by Gottschalk et al.

We again would like to thank Patrick Rafter and the anonymous reviewer for their evaluation of our manuscript. We have not made further changes to the manuscript in light of their comments below.

Reviewers' comments:

Reviewer #1 (Remarks to the Author):

The authors have completely addressed my comments / suggestions and the final manuscript will be an excellent and useful addition to the scientific literature.

Reviewer #2 (Remarks to the Author):

I am happy with the response of the authors to my query, and look forward to seeing the work published.